# Omnibus Modeling of *Listeria monocytogenes* Growth Rates at Low Temperatures

**DOI:** 10.3390/foods10051099

**Published:** 2021-05-15

**Authors:** Vincenzo Pennone, Ursula-Gonzales Barron, Kevin Hunt, Vasco Cadavez, Olivia McAuliffe, Francis Butler

**Affiliations:** 1Teagasc Food Research Centre, Moorepark, Fermoy, P61 C996 Co Cork, Ireland; vincenzopennone@hotmail.it (V.P.); Olivia.McAuliffe@teagasc.ie (O.M.); 2Centro de Investigação de Montanha (CIMO), Instituto Politécnico de Bragança, Campus de Santa Apolónia, 5300-253 Bragança, Portugal; ubarron@ipb.pt (U.G.-B.); vcadavez@ipb.pt (V.C.); 3UCD School of Biosystems and Food Engineering, University College Dublin, Belfield, Dublin 4, Ireland; kevin.hunt@ucd.ie

**Keywords:** omnibus modeling, *Listeria monocytogenes*, predictive microbiology, growth models, Huang model

## Abstract

*Listeria monocytogenes* is a pathogen of considerable public health importance with a high case fatality. *L. monocytogenes* can grow at refrigeration temperatures and is of particular concern for ready-to-eat foods that require refrigeration. There is substantial interest in conducting and modeling shelf-life studies on *L. monocytogenes*, especially relating to storage temperature. Growth model parameters are generally estimated from constant-temperature growth experiments. Traditionally, first-order and second-order modeling (or primary and secondary) of growth data has been done sequentially. However, omnibus modeling, using a mixed-effects nonlinear regression approach, can model a full dataset covering all experimental conditions in one step. This study compared omnibus modeling to conventional sequential first-order/second-order modeling of growth data for five strains of *L. monocytogenes*. The omnibus model coupled a Huang primary model for growth with secondary models for growth rate and lag phase duration. First-order modeling indicated there were small significant differences in growth rate depending on the strain at all temperatures. Omnibus modeling indicated smaller differences. Overall, there was broad agreement between the estimates of growth rate obtained by the first-order and omnibus modeling. Through an appropriate choice of fixed and random effects incorporated in the omnibus model, potential errors in a dataset from one environmental condition can be identified and explored.

## 1. Introduction

*Listeria monocytogenes* is a facultatively anaerobic Gram-positive bacterium which is pathogenic for humans and is of considerable public health importance. Within the European Union, EFSA [1] reported 2549 human cases of listeriosis (0.47 cases per 100,000 population) in 2018 with a statistically significant increasing trend from 2009–2018. While the number of cases per 100,000 population was lower than for other food-borne pathogens, EFSA reported that the case fatality was high (15.6%) and concluded that listeriosis is one of the most serious food-borne diseases [1]. *L. monocytogenes* is psychrotrophic; thus, it can grow at refrigeration temperatures. For this reason, *L. monocytogenes* is of particular concern for ready-to-eat (RTE) foods that require refrigerated storage. There has been substantial activity within the European Union and elsewhere to provide guidance on how to conduct laboratory shelf-life studies on *L. monocytogenes* in RTE foods to assure product safety and to conform to regulatory requirements. Recently, several guidance documents have emerged outlining the conduct of challenge studies relating to *L. monocytogenes* in RTE foods. The EC/DG SANCO guidance [2] documents a decision tree approach for the steps of shelf-life studies in order to investigate the growth of *L. monocytogenes* in the product. More recently, an EURL Lm Technical Guidance Document [3] details challenge tests and durability studies related to the growth of *L. monocytogenes* in RTE foods. In addition, an ISO standard—ISO 20976-1 [4]—provides further guidance on the conduct of challenge tests for any microorganism of concern. Other guidance documents are also available [5,6].

Growth kinetic parameters are generally estimated from constant-temperature growth experiments carried out during challenge testing using a primary growth model, where microbial counts are modeled as a function of time. Many primary models exist including the commonly used modified Gompertz model [7], the Baranyi model [8], the Buchanan model [9], and the Huang model [10]. Where appropriate, secondary models may also be applied to describe the effects of environmental factors such as temperature, water activity, and pH on the primary model parameters such as growth rate or lag time. A classic example of a secondary model is the Ratkowsky square-root model [11], which describes how growth rate varies with temperature.

Traditionally, first-order and second-order modeling has been done sequentially. Typically, a primary model is first fitted to a series of datasets on growth over time, carried out at constant temperature. The estimated parameters derived from the primary models are then subsequently used to fit to a secondary model describing, for example, the effect of temperature on growth rate. However, using a mixed-effects nonlinear regression approach, a full dataset covering all experimental conditions can be modeled at once. This type of modeling that fits the primary and secondary models at the same time is known as omnibus or global modeling [12,13]. Omnibus modeling is more advantageous than the classical two-step modeling because there is no loss of information related to the uncertainty of the kinetic parameters of the primary model, and random effects can accommodate the variability in parameters that cannot be explained by the environmental conditions [14]. The objective of this work was to compare omnibus modeling to a conventional sequential first-order/second-order modeling for growth studies carried out between 4 and 12 °C for five strains of *L. monocytogenes*. Previous studies demonstrated the strain-to-strain variability in growth rate for *L. monocytogenes* [15,16]. There is also evidence that the variability is higher at refrigerated temperatures of 4–7 °C [15,17,18].

## 2. Materials and Methods

### 2.1. Listeria monocytogenes Strains

Five *L. monocytogenes* strains were used for this study. The strains were chosen to be representative of ongoing challenge studies of *L. monocytogenes* in a range of dairy and seafood matrices. Three strains, 12MOB079LM (dairy source), 12MOB099LM (seafood), and 12MOB104LM (seafood), were selected from the EURL *Lm* strain collection [3]. Isolate 954 (dairy) was from the Teagasc *L. monocytogenes* collection (Teagasc, Moorepark, Co Cork, Ireland) and isolate 1513COB874 (seafood) was obtained from National University Ireland, Galway. The isolates were stored at −20 °C in 20% glycerol until required.

### 2.2. Growth Determination

As required, each strain was plated on Agar *Listeria* according to Ottaviani and Agosti (OAA—Ottaviani Agosti ready-to-use agar plates, BioMérieux, Marcy l’Etoile, France), and the plates were incubated for 48 h at 37 °C. One single colony of each isolate was resuspended in 10 mL of Brain Hearth Infusion broth (BHI, Merck, Ireland) and incubated at 37 °C for 18 h. Serial dilutions were performed with saline solution (NaCl 0.85%), and 100 µL of the final dilution was added to 10 mL of BHI broth, to ensure an inoculum concentration of nominally 1000 CFU/mL. The inoculum concentration was assessed by plate counting. Each BHI tube was incubated at 4.5 °C, 7.8 °C, or 12.0 °C. At each timepoint, serial dilutions were prepared, and 10 µL of each dilution was pipetted onto BHI agar. The plates were incubated at 37 °C for 18 h, and the colonies were counted for quantification. Each isolate was cultured in three biological replicates (starting each time with a new inoculum from the original isolate) for each temperature condition, and each dilution was plated in duplicate.

### 2.3. First-Order Growth Rate Modeling

To determine growth rate, the growth of *L. monocytogenes* at constant temperature was modeled using a three-phase Huang model [10]. This model is an adaption of a standard logistic growth curve to include an initial lag phase. The model uses four parameters to predict the CFU population at time *t* (*Nt*): the lag time (lag), unit h, the initial population (*N*0), the maximum growth rate (*μ*_max_), unit h^−1^, and the maximum population size (*N*_max_). α is the lag phase transition coefficient. Huang [10] reported that the value of α was not affected by bacteria type or temperature and determined it to be approximately 4, which is the value used in this study.
(1)$$\ln\left(Nt\right)=\ln\left(N0\right)+\ln\left(N_{\max}\right)-e^{\ln\left(N0\right)}+\left(e^{\ln(N_{\max})}-e^{\ln\left(N0\right)}\right)e^{-\mu_{\max}*\beta \left(t\right)}.$$
(2)$$\beta \left(t\right)=t+\frac{1}{\alpha }\ln\left(1+\frac{e^{-\alpha *\left(t-lag\right)}}{e^{\alpha *lag}}\right).$$

The model (Equation (1)) was fitted separately for each strain, temperature condition, and biological replicate using the package *nlme* in the statistical programming language R (R Core Group, [19]). For a small number of experiments at 7.8 °C, full stationary growth conditions were not achieved. In these cases, a simplified two-phase version [10] was fitted instead. This model reduces the parameters of interest to a lag phase (lag) and an exponential growth rate (*μ*_max_), units h and h^−1^, respectively (Equation (3)). For the reduced model, Huang [10] assigned a value of 25 to *α*, which is also the value used in this study.
(3)$$\ln Nt=\ln N0+\mu_{\max}*\left(t+\frac{1}{\alpha }*\ln\left(\frac{1+e^{-\alpha \left(t-lag\right)}}{1+e^{\alpha *lag}}\right)\right).$$

Differences between the growth rate parameter for each strain were compared using one-way analysis of variance at each temperature condition. If significance was determined, pairwise comparisons were made using a Tukey test. ANOVA assumptions were tested by residual analysis. Normality of residuals was assessed with a Shapiro–Wilk test, and homogeneity of variances was determined by a Levene test. All statistical analyses were carried out in R version 3.6.3 [19], using R Studio version 1.0.136.

### 2.4. Omnibus Modeling of Growth Curves

The growth of each of the *L. monocytogenes* strains was described by an omnibus model that coupled a primary model for growth to secondary models for growth rate and lag phase duration.

The omnibus model of the form,
(4)$$Y_{ij}=Y_{0j}+Y_{\max j}-\ln\left\{\exp\left(Y_{0j}\right)+\left(\exp(Y_{\max j}\right)-\exp\left(Y_{0j}\right))\times \exp\left(-\mu_{\max j}\times B_{ij}\right)\right\}+\varepsilon_{ij},$$
(5)$$B_{ij}=t_{i}+\frac{1}{\alpha }\ln\frac{1+\exp\left(-\alpha \times \left(t_{i}-\lambda_{j}\right)\right)}{1+\exp\left(\alpha \times \lambda_{j}\right)},$$
(6)$$Y_{0j}=Y_{0}+u_{j},$$
(7)$$\frac{1}{\sqrt{\lambda_{j}}}=\gamma_{0}+\gamma_{1}Temp_{j}^{2},$$
(8)$$\sqrt{\mu_{\max j}}=\left(\beta_{0}+v_{j}\right)+\beta_{1}Temp_{j},$$
(9)$$Y_{\max j}=Y_{\max}+w_{j},$$
is essentially a multilevel model, and it was adjusted to each of the five *L. monocytogenes* strain datasets, as a nonlinear mixed-effects regression.

The primary model for growth chosen was the Huang full model [10] (Equation (4)), whereas the secondary models were simple data-driven polynomial models that were defined before omnibus modeling. The goodness of fit of the secondary models for the lag phase duration (Equation (6)) and maximum growth rate (Equation (7)) was assessed by simple graphs of normality of residuals, predicted values versus observations, and residuals versus fitted values. More complex secondary models could not be considered since the number of temperature levels was only three.

The microbial concentration (ln CFU·mL^−1^) of *L. monocytogenes* measured at time *i* when subjected to the environmental condition *j* is represented by *Y_ij_*. *Y*_0*j*_ is the initial microbial concentration (ln CFU·mL^−1^) in a given environmental condition *j*. In this case, the environmental condition *j* is defined by the temperature of incubation *Temp* as a class variable. Since inoculum size was not exactly the same for every run, the mean initial microbial concentration *Y*_0_ was set to take in random effects *u_j_* that varied from condition to condition. *Y*_max_ denotes the maximum microbial concentration (ln CFU/mL), which is also affected by random deviations *w_j_* due to condition *j*. The microbial growth rate (ln CFU·h^−1^) and lag phase duration (h) of *L. monocytogenes*, belonging to the environmental condition *j*, are represented by *μ*_maxj_ and *λ_j_*, respectively. Nonetheless, to describe these two kinetic parameters as a function of temperature (*Temp*), they underwent transformations that were in advance proven to reduce heteroscedasticity. A square-root transformation was used for the microbial growth (√(*μ*_maxj_)), whereas a reciprocal square-root transformation was applied to the lag phase duration (1/√(*λ_j_*)). *γ*_1_ is the quadratic effect of temperature on 1/√(*λ_j_*), whereas *γ*_0_ is the intercept of this function. Residual or unexplained variability in √(*μ*_maxj_) was extracted by the random effects *v_j_* as realizations of the environmental condition *j*. Unlike the expression for the transformed maximum growth rate, which presented a stochastic component to account for the unexplained variability, the transformed lag phase duration (1/√(*λ_j_*)) could be estimated only by the fixed effects *γ*_0_ of the intercept and *γ*_1_ of the temperature. No residual variability term was added to 1/√(*λ_j_*) since the standard deviation of the random effects was consistently shown to be nonsignificant in the five omnibus models (i.e., belonging to the five *L. monocytogenes* strains). The random effects *u_j_*, *v_j_*, and *w_j_* were assumed to be correlated following a multi-normal distribution with mean zero and variance matrix $\left|\begin{array}{ccc} s_{u}^{2} & & \\ S_{u}S_{v} & s_{v}^{2} & \\ S_{u}S_{w} & S_{v}S_{w} & s_{w}^{2} \end{array}\right|$. Coefficients of correlations between random effects were then calculated. The residuals *ε_ij_* were assumed to follow a normal distribution with mean zero and standard deviation *s*. In addition to the parameter estimates and the variance components, the correlation coefficients between observations and fitted values (*R_obs-fit_*) and between fitted values and residuals (*R_fit-residuals_*) were calculated for every omnibus model, for assessment of goodness of fit.

### 2.5. Omnibus Modeling of the Growth of L. monocytogenes for the Five Strains

An overarching omnibus model was used to describe the growth of the five separate *L. monocytogenes* strains.
(10)$$Y_{is\left(j\right)}=Y_{0s\left(j\right)}+Y_{\max\;s\left(j\right)}-\ln\left\{\exp\left(Y_{0\;s\left(j\right)}\right)+\left(\exp(Y_{\max\;s\left(j\right)}\right)-\exp\left(Y_{0\;s\left(j\right)}\right))\times \exp\left(-\mu_{\max s\left(j\right)}\times B_{is\left(j\right)}\right)\right\}+\varepsilon_{is\left(j\right)}.$$
(11)$$B_{is\left(j\right)}=t_{i}+\frac{1}{4}\ln\frac{1+\exp\left(-4\times \left(t_{i}-\lambda_{s\left(j\right)}\right)\right)}{1+\exp\left(4\times \lambda_{s\left(j\right)}\right)}.$$
(12)$$Y_{0\;s\left(j\right)}=Y_{0}+u_{s}+u_{s\left(j\right)}.$$
(13)$$\frac{1}{\sqrt{\lambda_{s\left(j\right)}}}=\left(\gamma_{0}+v_{s}+v_{s\left(j\right)}\right)+\gamma_{1}Temp_{j}^{2}.$$
(14)$$\sqrt{\mu_{\max\;s\left(j\right)}}=(\beta_{0}+w_{s}+w_{s\left(j\right)})+\beta_{1}Temp_{j}.$$
(15)$$Y_{\max s(j)}=Y_{\max}+z_{s}+z_{s\left(j\right)}.$$

This general omnibus model presented the same fixed-effects architecture as the strain-specific omnibus model (Equation (4)), yet with a different clustering structure that nested environmental conditions *j* within strain s. Thus, Equation (9) was fitted to the entire dataset, whereby the observations $Y_{is\left(j\right)}$ were now defined as the microbial concentration (ln CFU·mL^−1^) of *L. monocytogenes* strain *s* measured at time *i* when subjected to the environmental condition *j*. The variables $Y_{0\;s\left(j\right)}$, $1/\sqrt{\lambda_{s\left(j\right)}}$, $\sqrt{\mu_{\max\;s\left(j\right)}}$, and $Y_{\max s(j)}$ were allowed to take in the nested random shifts $u_{s}+u_{s\left(j\right)}$, $v_{s}+v_{s\left(j\right)}$, $w_{s}+w_{s\left(j\right)}$, and $z_{s}+z_{s\left(j\right)}$, respectively, as realizations of the strain *s* and the environmental condition *j* nested in strain *s*. With this error structure arrangement, it is possible to separate the residual variability in $Y_{is\left(j\right)}$ due to the experimental condition from the actual inter-strain variability. The random effects $u_{s}$, $v_{s}$, $w_{s}$, and $z_{s}$ were assumed to be correlated and distributed as a multi-normal distribution with mean zero and the following covariance matrix:(16)$$\left|\begin{array}{cccc} s_{u\;s}^{2} & & & \\ S_{u\;s}S_{v\;s} & s_{v\;s}^{2} & & \\ S_{u\;s}S_{w\;s} & S_{v\;s}S_{w\;s} & s_{w\;s}^{2} & \\ S_{u\;s}S_{z\;s} & S_{v\;s}S_{z\;s} & S_{w\;s}S_{z\;s} & s_{z\;s}^{2} \end{array}\right|$$

Likewise, the nested random effects $u_{s\left(j\right)}$, $v_{s\left(j\right)}$, $w_{s\left(j\right)}$, and $z_{s\left(j\right)}$ were assumed to be correlated and normally distributed with mean zero and the following covariance matrix:(17)$$\left|\begin{array}{cccc} s_{u\;s(j)}^{2} & & & \\ S_{u\;s(j)}S_{v\;s(j)} & s_{v\;s(j)}^{2} & & \\ S_{u\;s(j)}S_{w\;s(j)} & S_{v\;s(j)}S_{w\;s(j)} & s_{w\;s(j)}^{2} & \\ S_{u\;s(j)}S_{z\;s(j)} & S_{v\;s(j)}S_{z\;s(j)} & S_{w\;s(j)}S_{z\;s(j)} & s_{z\;s(j)}^{2} \end{array}\right|$$

Coefficients of correlation between the nested random effects were calculated from the two matrices. The residuals $\varepsilon_{is\left(j\right)}$ were assumed to follow a normal distribution with mean zero and standard deviation *s*.

### 2.6. Validation of the Omnibus Model

The overarching omnibus model shown in Equation (5) was fitted to the datasets from two of the temperatures, 4.5 and 12 °C, and used to predict the growth at 7.8 °C for model validation. The predictive capacity of the omnibus model was assessed by means of graphs of growth observations versus predicted growth curves for the five strains. In addition to the visual appraisal, the accuracy factor (Af) and bias factor (Bf) of the growth predictions were jointly calculated for all the growth curves [20]. All the nonlinear mixed-effects regression models were fitted using the *nlme* function from the *nlme* package [21] implemented in R Studio version 1.0.136 with R version 3.6.3 [19].

## 3. Results

### 3.1. Growth Curves

Figure 1 shows the representative growth curves for two of the strains at the three experimental temperatures. Superimposed on the experimental data points are the three-phase Huang model predictions. A period of exponential growth was observed in all strains at the three temperatures. As expected, as temperature increased, the lag time became shorter and the growth rate during the exponential phase became higher. For a small number of experiments at 7.8 °C, full stationary growth conditions were not achieved. However, an estimate of the maximum concentration was normally possible from the models.

### 3.2. First-Order Growth Rate Modeling

Table 1 summarizes the growth rate parameter estimates for each strain at the recorded temperature. There were significant differences in growth rate depending on the strain at all three temperatures. At 4.5 °C, the mean growth rate varied between 0.0428 and 0.0476 h^−1^. On the other hand, at 12 °C, the growth rate varied between 0.145 and 0.187 h^−1^. The secondary effect of temperature on growth rate was compared among strains with multiple linear regression. Maximum growth rates were fitted with a square-root transformation, using the predictors temperature, strain, and their interaction. Figure 2 shows the growth rate values for each *L. monocytogenes* strain at each temperature. Multiple linear regression showed no significant difference between the intercept for any strain and only one significant (*p* < 0.05) difference in slope, for strain 1513COB874. Figure 2 also shows that, at 12 °C, there was more variability in the growth rate compared to the lower temperatures.

### 3.3. Omnibus Modeling of Growth Curves

Table 2, Table 3, Table 4 and Table 5 set out the omnibus model parameters for the five strains based on the complete dataset at the three temperatures. Estimates of the fixed-effect parameters, *Y*_0_ (initial concentration (ln CFU·mL^−1^)) were similar (~7.3 ln; 3.2 log_10_) across the strains, with the exception of strain 12MOB104LM (6.7 ln; 2.9 log_10_). Likewise, estimates of *Y*_max_ (final concentration (ln CFU·mL^−1^)) were similar across the five strains (~21.4 ln; 9.3 log_10_). Estimates of the predictors of √*µ*_max_, *β*_0_ and *β*_1_, were also similar for the five strains, with the exception of strain 1513COB874. This outcome is consistent with what was reported from the first-order modeling and shown in Figure 2. The standard error reported for the estimates of the predictors of √*µ*_max_ were all low, indicating good consistency among replicates.

With the exception of *L. monocytogenes* strain 12MOB079LM (Table 3), the between-condition variability in *Y*_0_ (*s_u_*), √*μ*_max_ (*s_v_*), and *Y*_max_ (*s_w_*) exhibited a similar trend for all strains. The variability in *Y*_max_ ranged between 0.284 and 0.848 ln CFU·mL^−1^ (see *s_w_* in Table 2, Table 4, Table 5 and Table 6) and was consistently higher than the variability in *Y*_0_, which fell between 0.021 and 0.442 ln CFU·mL^−1^ (see *s_u_* in Table 2, Table 4, Table 5 and Table 6). Since the omnibus models suggested that these two variabilities were not correlated (*r* = −0.582, 0.006, −0.165, −0.091, and −0.288 for all strains), it is reasonable to conclude that the inoculum concentration (related to *Y*_0_) did not exert any effect on the maximum concentration reached in every experiment. Therefore, the higher variability observed in the maximum concentration may be linked to the absence of an observed stationary phase in some of the 7.8 °C data. This uncertainty increases the error in *Y*_max_. In comparison to the variability in *Y*_0_ (*s_u_*) and *Y*_max_ (*s_w_*), the variability in √*μ*_max_ was lower in all strains (*s_v_* ranged from virtually zero to 0.017 in Table 2, Table 3, Table 4, Table 5 and Table 6), which indicates that temperature was able to explain most of the variability in √*μ*_max_, with low error associated with the estimation of √*μ*_max_. This fact further indicates that the absence of a stationary phase in the 7.8 °C data did not affect the accurate estimation of √*μ*_max_.

### 3.4. Omnibus Modeling of the Growth of L. monocytogenes for the Five Strains

Table 7 sets out the parameters from the overarching omnibus model used to describe the growth of the five *L. monocytogenes* strains. With the error structure arrangement, set up in Equation (5), it is possible to separate the residual variability in $Y_{is\left(j\right)}$ due to the experimental condition from the actual inter-strain variability. The values estimated for random effects due to strain, ranging between 8.5 × 10^−8^ for variability in lag phase and 5.7 × 10^−5^ for variability in maximum concentration (Table 7), clearly indicated that the variability among strains was negligible when assessed by omnibus modeling. In other words, strain did not appear to exert any effect on the microbial kinetic parameters.

However, most or nearly all variability in the kinetic parameters was as a result of the experimental conditions (temperature within strain). Again, the error in *Y*_max_ (0.450 ln CFU·mL^−1^) was slightly higher than the error in *Y*_0_ (0.305 ln CFU·mL^−1^) for the same reasons explained in the previous section The overarching omnibus model confirmed that the between-condition error associated with the estimation of 1/√*λ* was negligible (*s*_*v s*(*j*)_ = 8.2 × 10^−8^), while the error in estimating √*µ*_max_ was also low (*s*_*w s*(*j*)_ = 0.017; Table 7). This demonstrates both (i) that the linear predictors of 1/√*λ* and √*µ*_max_ defined in Equation (5) are adequate for describing the data and (ii) that the experiments were undertaken with good repeatability.

Using the estimated parameters in Table 7, it is possible to calculate maximum growth rates (*μ*_max_) at 4.5, 7.8, and 12.0 °C for the five strains (Table 8). There was very little difference in growth rates among strains at 4.5 °C. The variability among strains was largest at 12.0 °C, with strain 1513COB874 having the largest estimated strain rate. This is consistent with what was observed in Figure 2. In Figure 3, growth curves for the five strains of *L. monocytogenes* at 7.8 °C as predicted by the overarching omnibus model are superimposed on the count observations using the 7.8 °C data as a separate validation dataset. Agreement between the experimental data and predicted growth curves was very good in all cases (pooled Af = 1.173 and pooled Bf = 1.094).

## 4. Discussion

Analysis of variance showed that there were some significant differences in growth rates among strains, which increased with temperature. At 4.5 °C, the largest difference was 11%, but this increased to 29% at 12 °C. To put this into perspective, the difference in reported growth rates [22] for the three the EURL *Lm* strains included in this study was 6% at 8 °C (Table 9). Moreover, Silva et al. [23] developed a web-based SHINY app which reports a meta-analysis estimate of growth rate for *L. monocytogenes* of 0.063 h^−1^ with a 95% confidence interval between 0.0517 and 0.0755 h^−1^ at 8 °C. The relatively large confidence intervals for the estimates of the growth rates in the present study (Table 8), determined from the omnibus modeling, indicate possibly small practical differences between strains for the growth rates estimated at a given temperature. Comparing Table 1 and Table 8, there was broad agreement between the outcomes of the first-order and the omnibus modeling of growth rates across the five strains and three temperatures. The largest differences were at 7.8 °C, where the average differences between the two estimates was 17%. Again, these differences need to be judged within the context of the large confidence intervals for omnibus-derived growth rates already highlighted.

Three of the strains were selected from the EURL *Lm* strain collection where reported growth rates have previously been reported [22]. In Table 9, the growth rate estimates from the omnibus modeling are compared to the EURL *Lm*-reported values [22] and to estimates derived from the publicly available COMBASE [24] predictions for *L. monocytogenes*, as well as predictions reported [23] using the web-based SHINY app. By necessity, there were small differences in the environmental conditions of temperature, pH, and Aw. The reported growth rates for the EURL *Lm* strains [22] were 10–19% higher than the values determined in the current study. However, the reported EURL *Lm* growth rates were determined by optical density measurements of cell growth against plate counts as used in the present study. Silva et al. [23] found using meta-analysis that growth rates determined by optical density measurements were normally higher than growth rates determined using conventional plating techniques. It is noteworthy that the two strains 12MOB079LM and 12MOB099LM are recommended for inclusion in low-temperature challenge tests [3] because of their high growth rates at low temperatures. This would partly explain why the current growth rate estimates for these two strains are higher than those predicted by the SHINY app [23], which represents a meta-analysis-derived average of many growth rate studies.

There have been relatively few published papers relating to the application of omnibus modeling to the growth of *L. monocytogenes*. To illustrate his new growth model, Huang [25] applied a primary model to the growth of *L. monocytogenes* in broth and frankfurters but did not employ omnibus modeling. Liu et al. [26] used omnibus modeling to describe the growth of *L. monocytogenes* in ready-to-eat braised beef. Other studies have also reported on the omnibus modeling approach [12,27,28]. The current work demonstrated a number of advantages of the omnibus modeling approach. As the model is informed by all of the datasets obtained at different temperatures, there is no loss of information, and potential systematic errors in a dataset from one environmental condition can be identified and explored through an appropriate choice of fixed and random effects incorporated in the model. It is noteworthy that the first-order modeling predicted varying final-stage concentrations depending on temperature (Figure 1), whereas the omnibus modeling approach predicted an essentially similar final concentration irrespective of the temperature. Another benefit is the ability to interpolate across environmental conditions, such as temperature values, and to potentially compensate for incomplete datasets for one environmental condition. It is normal that, in comparison to the two-step approach, omnibus modeling produces wider confidence intervals for estimates of microbial concentrations or kinetic parameters, since, as there is no information loss and an error structure can be defined, the variance of the random effects extracted adds to the total variance associated to a prediction. Therefore, a higher number of significant random effects leads to wider confidence intervals.

In conclusion, the omnibus modeling approach represents a useful complementary approach to the classical methodology of sequential first-order and second-order modeling to further explore and gain insights from experimental growth data.

## Figures and Tables

**Figure 1 foods-10-01099-f001:**
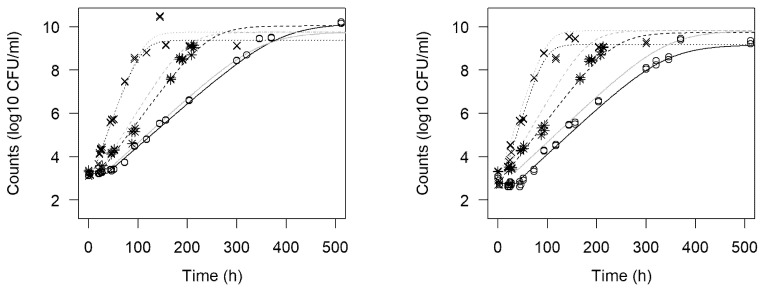
Representative experimental growth curves (one replicate) for *L. monocytogenes* strains 12MOB079LM (**left**) and 12MOB099LM (**right**) at the three measured temperatures (o 4.5 °C, ✷ 7.8 °C, ×12 °C) together with Huang model predictions derived from first-order modeling (darker lines) and omnibus modeling (lighter lines).

**Figure 2 foods-10-01099-f002:**
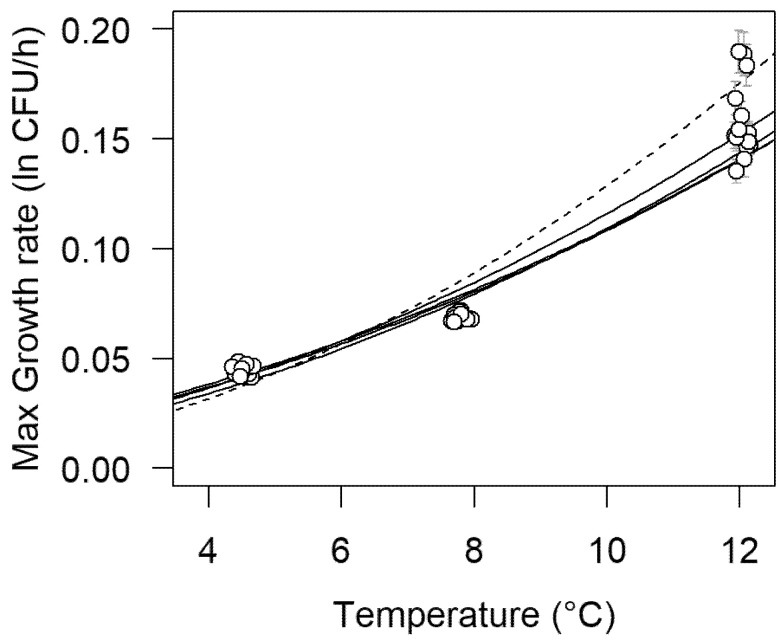
Maximum growth rate values for each *L. monocytogenes* strain (all replicates plotted) at each temperature (data points stretched at each temperature for clarity). Square-root transformed linear regressions are also plotted. The dotted line represents strain 1513COB874, whose transformed slope was the only significant difference observable (*p* < 0.05). A small random horizontal shift was added to the data points to improve their visibility.

**Figure 3 foods-10-01099-f003:**
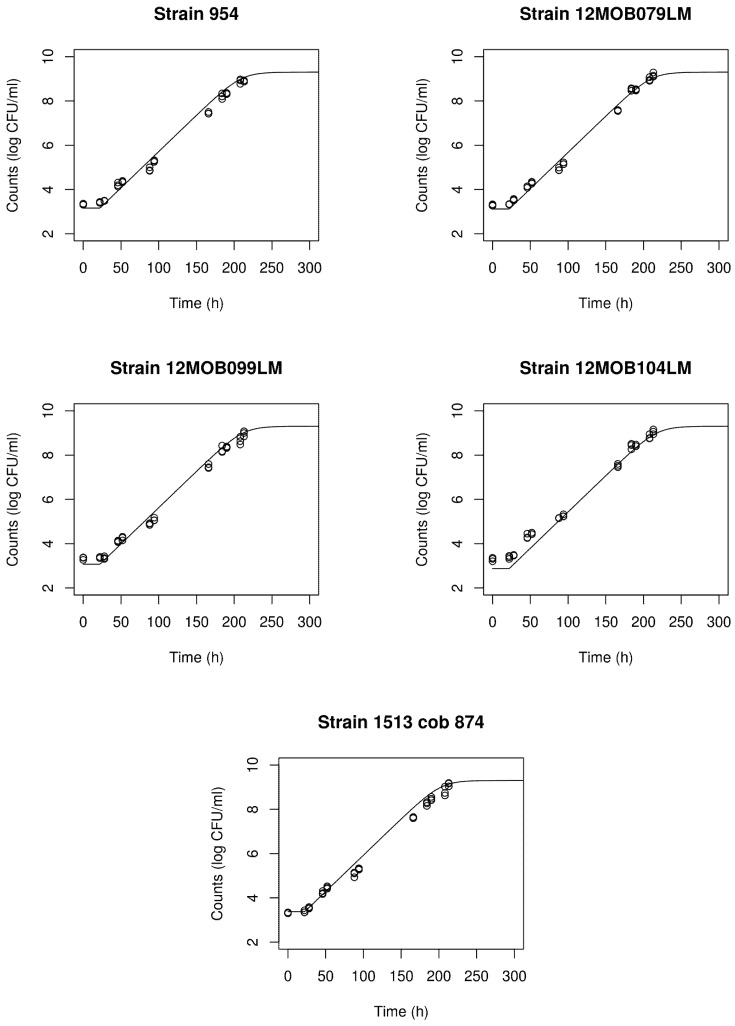
Growth curves for the five strains of *L. monocytogenes* stored at 7.8 °C, predicted by the overarching omnibus model against count observations using the separate validation dataset (pooled Af = 1.173 and pooled Bf = 1.094).

**Table 1 foods-10-01099-t001:** Growth rate parameter estimates (STANDARD deviation of the three replicates in parenthesis) for each strain of *L. monocytogenes* and experimental temperature condition obtained by first-order modeling. Averages within a column followed by a common superscript letter are not significantly different (*p* < 0.05).

Strain	4.5 °C	Growth Rate (h^−1^) 7.8 °C	12.0 °C
954	0.043 ^b^ (0.00091)	0.068 ^b^ (0.00045)	0.15 ^bc^ (0.0027)
12MOB079LM	0.046 ^a^ (0.00038)	0.071 ^a^ (0.00069)	0.15 ^c^ (0.0040)
12MOB099LM	0.046 ^ab^ (0.00098)	0.069 ^ab^ (0.00071)	0.15 ^bc^ (0.0093)
12MOB104LM	0.048 ^a^ (0.0011)	0.068 ^b^ (0.00017)	0.16 ^b^ (0.0070)
1513COB874	0.045 ^ab^ (0.0022)	0.068 ^b^ (0.0016)	0.19 ^a^ (0.0034)

**Table 2 foods-10-01099-t002:** Fixed- and random-effects estimates of the omnibus model describing the concentration (ln CFU/mL) of *L. monocytogenes* strain 954. Goodness-of-fit measures include coefficient of correlation of observed versus fitted values (R_obs-fit_) and coefficient of correlation of fitted values versus residuals (R_fit-residuals_).

Parameters	Mean	Standard Error	Pr > |t|	Other Analysis
Fixed effects				
Y_0_ (ln CFU·mL^−1^)	7.280	0.205	<0.0001	
Y_max_ (ln CFU·mL^−1^)	21.48	0.475	<0.0001	
Predictor of √µ_max_				
β_0_ (Intercept) (h^−0.5^)	0.108	0.004	<0.0001	
β_1_ (Temp) (h^−0.5^·°C^−1^)	0.021	0.001	<0.0001	
Predictor of 1/√λ				
γ_0_ (Intercept) (h^−0.5^)	0.145	0.009	<0.0001	R_obs-fit_ = 0.997
γ_1_ (Temp²) (h^−0.5^·°C^−2^)	0.0017	0.0003	<0.0001	R_fit-residuals_ = 0.008
Random effects (condition)		Correlation matrix		
s_u_ (Y_0_) (ln CFU·mL^−1^)	0.302	s_u_ (Y_0_)	s_w_ (Y_max_)	
s_v_ (β_0_) (h^−0.5^)	1.7 × 10^−9^	0.725		
s_w_ (Y_max_) (ln CFU·mL^−1^)	0.774	−0.582	−0.284	
s (residual)	0.507			

**Table 3 foods-10-01099-t003:** Fixed- and random-effects estimates of the omnibus model describing the concentration (ln CFU/mL) of *L. monocytogenes* strain 12MOB079LM. Goodness-of-fit measures include coefficient of correlation of observed versus fitted values (R_obs-fit_) and coefficient of correlation of fitted values versus residuals (R_fit-residuals_).

Parameters	Mean	Standard Error	Pr > |t|	Other Analysis
Fixed effects				
Y_0_ (ln CFU·mL^−1^)	7.318	0.107	<0.0001	
Y_max_ (ln CFU·mL^−1^)	21.86	0.135	<0.0001	
Predictor of √µ_max_				
β_0_ (Intercept) (h^−0.5^)	0.111	0.016	<0.0001	
β_1_ (Temp) (h^−0.5^·°C^−1^)	0.022	0.002	<0.0001	
Predictor of 1/√λ				
γ_0_ (Intercept) (h^−0.5^)	0.135	0.010	<0.0001	R_obs-fit_ = 0.995
γ_1_ (Temp²) (h^−0.5^·°C^−2^)	0.001	0.0002	<0.0001	R_fit-residuals_ = 0.002
Random effects (condition)		Correlation		
s_u_ (Y_0_) (ln CFU·mL^−1^)	0.021	s_u_ (Y_0_)	s_w_ (Y_max_)	
s_v_ (β_0_) (h^−0.5^)	0.009	0.177		
s_w_ (Y_max_) (ln CFU·mL^−1^)	3.2 × 10^−7^	0.006	0.030	
s (residual)	0.572			

**Table 4 foods-10-01099-t004:** Fixed- and random-effects estimates of the omnibus model describing the concentration (ln CFU/mL) of *L. monocytogenes* strain 12MOB099LM. Goodness-of-fit measures include coefficient of correlation of observed versus fitted values (R_obs-fit_) and coefficient of correlation of fitted values versus residuals (R_fit-residuals_).

Parameters	Mean	Standard Error	Pr > |t|	Other Analysis
Fixed effects				
Y_0_ (ln CFU·mL^−1^)	7.420	0.170	<0.0001	
Y_max_ (ln CFU·mL^−1^)	21.09	0.208	<0.0001	
Predictor of √µ_max_				
β_0_ (Intercept) (h^−0.5^)	0.101	0.012	<0.0001	
β_1_ (Temp) (h^−0.5^·°C^−1^)	0.023	0.001	<0.0001	
Predictor of 1/√λ				
γ_0_ (Intercept) (h^−0.5^)	0.129	0.009	<0.0001	R_obs-fit_ = 0.996
γ_1_ (Temp²) (h^−0.5^·°C^−2^)	0.001	0.0002	<0.0001	R_fit-residuals_ = 0.001
Random effects (condition)		Correlation matrix		
s_u_ (Y_0_) (ln CFU·mL^−1^)	0.232	s_u_ (Y_0_)	s_w_ (Y_max_)	
s_v_ (β_0_) (h^−0.5^)	0.013	0.171		
s_w_ (Y_max_) (ln CFU·mL^−1^)	0.284	−0.165	0.051	
s (residual)	0.508			

**Table 5 foods-10-01099-t005:** Fixed- and random-effects estimates of the omnibus model describing the concentration (ln CFU/mL) of *L. monocytogenes* strain 12MOB104LM. Goodness-of-fit measures include coefficient of correlation of observed versus fitted values (R_obs-fit_) and coefficient of correlation of fitted values versus residuals (R_fit-residuals_).

Parameters	Mean	Standard Error	Pr > |t|	Other Analysis
Fixed effects				
Y_0_ (ln CFU·mL^−1^)	6.709	0.241	<0.0001	
Y_max_ (ln CFU·mL^−1^)	21.58	0.331	<0.0001	
Predictor of √µ_max_				
β_0_ (Intercept) (h^−0.5^)	0.097	0.012	<0.0001	
β_1_ (Temp) (h^−0.5^·°C^−1^)	0.024	0.001	<0.0001	
Predictor of 1/√λ				
γ_0_ (Intercept) (h^−0.5^)	0.129	0.011	<0.0001	R_obs-fit_ = 0.996
γ_1_ (Temp²) (h^−0.5^·°C^−2^)	0.002	0.0003	<0.0001	R_fit-residuals_ = 0.013
Random effects (condition)		Correlation matrix		
s_u_ (Y_0_) (ln CFU·mL^−1^)	0.373	s_u_ (Y_0_)	s_w_ (Y_max_)	
s_v_ (β_0_) (h^−0.5^)	0.017	0.016		
s_w_ (Y_max_) (ln CFU·mL^−1^)	0.518	−0.091	0.031	
s (residual)	0.495			

**Table 6 foods-10-01099-t006:** Fixed- and random-effects estimates of the omnibus model describing the concentration (ln CFU/mL) of *L. monocytogenes* strain 1513COB874. Goodness-of-fit measures include coefficient of correlation of observed versus fitted values (R_obs-fit_) and coefficient of correlation of fitted values versus residuals (R_fit-residuals_).

Parameters	Mean	Standard Error	Pr > |t|	Other Analysis
Fixed effects				
Y_0_ (ln CFU·mL^−1^)	7.336	0.294	<0.0001	
Y_max_ (ln CFU·mL^−1^)	21.05	0.518	<0.0001	
Predictor of √µ_max_				
β_0_ (Intercept) (h^−0.5^)	0.089	0.007	<0.0001	
β_1_ (Temp) (h^−0.5^·°C^−1^)	0.027	0.001	<0.0001	
Predictor of 1/√λ				
γ_0_ (Intercept) (h^−0.5^)	0.145	0.016	<0.0001	R_obs-fit_ = 0.993
γ_1_ (Temp²) (h^−0.5^·°C^−2^)	0.001	0.0003	<0.0001	R_fit-residuals_ = −0.003
Random effects (condition)		Correlation matrix		
s_u_ (Y_0_) (ln CFU·mL^−1^)	0.442	s_u_ (Y_0_)	s_w_ (Y_max_)	
s_v_ (β_0_) (h^−0.5^)	9.9 × 10^−9^	0.689		
s_w_ (Y_max_) (ln CFU·mL^−1^)	0.848	−0.288	−0.225	
s (residual)	0.667			

**Table 7 foods-10-01099-t007:** Fixed- and random-effects estimates of the overarching omnibus model describing the concentration (ln CFU·mL^−1^) of five strains of *L. monocytogenes*. Goodness-of-fit measures include coefficient of correlation of observed versus fitted values (R_obs-fit_) and coefficient of correlation of fitted values versus residuals (R_fit-residuals_).

Parameters	Mean	Standard Error	Pr > |t|	Other Analysis
Fixed effects				
Y_0_ (ln CFU·mL^−1^)	7.286	0.092	<0.0001	
Y_max_ (ln CFU·mL^−1^)	21.69	0.150	<0.0001	
Predictor of √µ_max_				
β_0_ (Intercept) (h^−0.5^)	0.098	0.013	<0.0001	
β_1_ (Temp) (h^−0.5^·°C^−1^)	0.024	0.001	<0.0001	
Predictor of 1/√λ				
γ_0_ (Intercept) (h^−0.5^)	0.144	0.005	<0.0001	R_obs-fit_ = 0.996
γ_1_ (Temp²) (h^−0.5^·°C^−2^)	0.0013	0.0001	<0.0001	R_fit-residuals_ = 0.006
Nested random effects				
Strain *s*		Correlation matrix		
s_u s_ (Y_0_) (ln CFU·mL^−1^)	3.2 × 10^−5^	s_v s_ (γ_0_)	s_w s_ (β_0_)	s_z s_ (Y_max_)
s_v s_ (γ_0_) (h^−0.5^)	8.5 × 10^−8^	−0.148		
s_w s_ (β_0_) (h^−0.5^)	2.8 × 10^−6^	0.360	−0.226	
s_z s_ (Y_max_) (ln CFU·mL^−1^)	5.7 × 10^−5^	0.245	−0.003	0.543
Condition *j* in Strain *s*				
s_u s(j)_ (Y_0_) (ln CFU·mL^−1^)	0.305	s_v s(j)_ (γ_0_)	s_w s(j)_ (β_0_)	s_z s(j)_ (Y_max_)
s_v s(j)_ (γ_0_) (h^−0.5^)	8.2 × 10^−8^	−0.763		
s_w s(j)_ (β_0_) (h^−0.5^)	0.017	−0.385	0.719	
s_z s(j)_ (Y_max_) (ln CFU·mL^−1^)	0.450	0.514	−0.511	−0.552
s (residual)	0.498			

**Table 8 foods-10-01099-t008:** Mean and 95% confidence interval (CI) of the microbial growth rate (μ_max_) at 4.5, 7.8, and 12.0 °C for the five strains of *L. monocytogenes*, as estimated from the omnibus model.

Temperature (°C)	Strain	Mean μ_max_ (h^−1^)	Low CI μ_max_ (h^−1^)	High CI μ_max_ (h^−1^)
4.5	954	0.0410	0.0364	0.0459
	12MOB079LM	0.0441	0.0304	0.0606
	12MOB099LM	0.0418	0.0321	0.0527
	12MOB104LM	0.0420	0.0324	0.0528
	1513COB874	0.0443	0.0377	0.0515
7.8	954	0.0738	0.0649	0.0835
	12MOB079LM	0.0799	0.0573	0.1068
	12MOB099LM	0.0787	0.0635	0.0951
	12MOB104LM	0.0807	0.0655	0.0973
	1513COB874	0.0897	0.0780	0.1025
12	954	0.1126	0.1049	0.1480
	12MOB079LM	0.1407	0.1017	0.1867
	12MOB099LM	0.1422	0.1180	0.1684
	12MOB104LM	0.1481	0.1234	0.1748
	1513COB874	0.1705	0.1490	0.1939

**Table 9 foods-10-01099-t009:** Comparison of estimated maximum growth rates (h^−1^)) at 7.8 °C for the EURL *Lm* strains used in the study with published results.

Study	Current	*EURL* [18]	Shiny [21]	Combase [19]
Environment				
Temperature (°C)	7.8	8	8	8
pH	7.2	7	7	7
Aw	0.997	0.98	0.98	0.98
Strain				
12MOB079LM	0.080	0.093		
12MOB099LM	0.079	0.094		
12MOB104LM	0.081	0.089		
Model (op. density)			0.086	
Model (cell count)			0.063	0.071

## Data Availability

The data presented in this study are available on request from the corresponding author.

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
