# Peer review of "Omnibus Modeling of Listeria monocytogenes Growth Rates at Low Temperatures"

_foods, 2021, doi:10.3390/foods10051099_

Round 1
Reviewer 1 Report
The manuscript describes the application of omnibus modelling to the growth of several Listeria monocytogenes strains at low temperatures. The research has been nicely conducted, the manuscript contains interesting results and these results are clearly explained.
Only the following minor remarks are raised:
1.-Although it may seem obvious, but parameters of equations 1, 2 and 3 are not defined in the text, so please, define them.
2.-Second order modelling is not described in the material and methods section, but it deserves a whole subsection.
3.-Figure 2 shows the maximum growth rate values in front of temperature, but authors state that they were fitted with a square root transformation. Why not plotting the square root of the growth rate in front of the temperature instead?
4.-Lines 360-361: It is stated that strain did not appear to exert any effect on the microbial kinetic parameters, referring to variability in lag phase and maximum concentration, but, for example, table 1 shows significant differences on the effect of temperature on growth rate between strains, so the sentence should be changed. Then, the statement can lead to misunderstanding and should be changed.
Author Response
The manuscript describes the application of omnibus modelling to the growth of several Listeria monocytogenes strains at low temperatures. The research has been nicely conducted, the manuscript contains interesting results and these results are clearly explained.
Response: Thank you for this positive response to this paper.
1.-Although it may seem obvious, but parameters of equations 1, 2 and 3 are not defined in the text, so please, define them.
Response: That was an omission and they have now been inserted in the paragraph starting line 121.
2.-Second order modelling is not described in the material and methods section, but it deserves a whole subsection.
Response: We have now better described the second order modelling. While we do not think that the secondary modelling requires a whole subsection, because these are simple linear equations of the transformed variables growth rate and lag phase duration. Nonetheless a section on secondary modelling and assessment of their goodness of fit has been added in the Subsection of Omnibus Modelling.
3.-Figure 2 shows the maximum growth rate values in front of temperature, but authors state that they were fitted with a square root transformation. Why not plotting the square root of the growth rate in front of the temperature instead?
Response: The maximum growth rate values were transformed back from square root values to give a better visualisation to the reader of how the values were varying with temperature.
4.-Lines 360-361: It is stated that strain did not appear to exert any effect on the microbial kinetic parameters, referring to variability in lag phase and maximum concentration, but, for example, table 1 shows significant differences on the effect of temperature on growth rate between strains, so the sentence should be changed. Then, the statement can lead to misunderstanding and should be changed.
Response: The values in table one were derived from the first order modelling which did show a small significant difference as indicated by the varying superscripts in Table one.. However the text in lines 360-361 refers specifically to the output of the overarching omnibus model. The differing outcomes from the first order modelling and the omnibus modelling is discussed in the discussion in the paragraph starting line 412. However additional text has been added to the caption of Table one and in line 363 to clearly distinguish the two modelling approaches.
Reviewer 2 Report
This is a well written manuscript documenting the application of omnibus method in predictive modeling. The manuscript is very easy to follow. The research is original and will add to the scientific literature. It can be published after minor revision. I made comments directly in the pdf. Please use it for revision.

Author Response
This is a well written manuscript documenting the application of omnibus method in predictive modeling. The manuscript is very easy to follow. The research is original and will add to the scientific literature. It can be published after minor revision. I made comments directly in the pdf. Please use it for revision.
Response: Thank you for the positive review, the minor editing details in peer-review-10935034.v1.pdf have all been attended to. The constants 4 and 25 have been replaced by a constant, α, in line with the original notation as used in the Huang paper (reference 10). Α can either be 4 or 25 depending on the model in line with reference 15.
With respect to the comment relating to the use of data coming from the SHINY app. This website indicates a difference determined by meta-analysis between values of the growth rate determined by plating or by optical density. As table 9 shows, this difference is between 0.063 and 0.086 h-1. This is comparable to the difference in growth rates reported by this study for the EURL strains and what was originally reported noting that this study used plate counts whereas the EURL reference reports optical density. We are of the view that this difference merits pointing out.
Reviewer 3 Report
In the manuscript entitled “Omnibus modelling of Listeria monocytogenes growth rates at low temperatures”, the authors apply a global approach to describe the growth kinetics of L. monocytogenes are refrigeration temperatures. Although the approach could be of interest as it enables to include biological variability, there are severe deficiencies in experimental design and model validation in this study. Moreover, some hypotheses in model building should be better reasoned, and there
Major comments
--------------------
## The number of strains chosen in the study is too small to properly describe between-strain variability
One of the advantages of multilevel models is the explicit definition of variance components to describe the variability in the microbial response. The authors make a link in several parts of the manuscript between their modelling approach and the study of variability (e.g. L79 or L209-211). It is important to highlight that this is an empirical model, whose quality is (at most) as good as the quality of the data used to build it. For instance, when describing a growth curve, an article would never be accepted if it had just a couple of time points per experiment. The same logic applies to studies on variability: their estimates can only be precise if the set of strains included in the study are representative of the actual population of Listeria strains.
I believe this is the reason the authors conclude there is no between-strain variability in growth (L360-361). This is a surprising result that goes against most of the scientific literature, which has observed differences in the growth characteristic at the between-strain level. This is most likely a result of the narrow experimental design (just 5 strains , whereas e.g. (Aryani et al., 2015) studied 20 strains). Therefore, the estimates of between-strain variability would be an underestimation. Considering that variability can be as relevant as kinetic parameters for shelf life estimation and risk assessment (specially for a microorganism with the ecology of L.monocytogenes), this model could not be used as a overarching model that generally describes the response of Listeria monocytogenes. Then, it could be used as just one model to describe the response of a single strain. But there are already thousands of growth curves published for L.monocytogenes at this temperature range.
## The selection of the method for model validation is surprising
In empirical models, model validation is done to assess whether the model is able to predict the microbial response at conditions different to the ones used to build the model (i.e. model fitting). For instance, a model fitted to data at 5 and 10ºC, can be validated using independent data at 7ºC to assess whether the model is valid for temperatures within 5 and 10ºC.
I find it surprising how validation is interpreted by the authors. They used as training set 2 of the replicates, using the third one as validation (L226-229). That is, they are using exactly the same conditions (storage time, storage temperature, strain, pre-incubation conditions…) for training and validation. From my understanding, this serves to validate that the model would be good at predicting a fourth replication of this experiment, nothing more. Then, how useful is this model for risk assessment?
## There are several hypotheses in model construction that are arguable
(listed in the minor comments below)
## The use of the term “omnibus” gives a misleading exaggeration of novelty
The authors use the term “omnibus” to define their approach, a term that is not broadly used in the field. This gives the impression that this approach is entirely novel. To my knowledge, it has only been used in the two articles referred by the authors (L74), which were written by the same authors.
However, the application of mixed effects models in predictive microbiology is not new. For instance, most meta-regression studies have applied this approach (e.g. (Jaloustre et al., 2012; Nunes Silva et al., 2020). It could be argued this method is more “advanced” than those, as it includes the errors at different levels. However, the methodology applied by the authors is similar to the one applied in several recent studies (Garre et al., 2020; van Boekel, 2020). It might be argued that the novelty comes from this methodology being adapted for growth, whereas previous studies applied it for inactivation (although this argument is quite weak in my opinion). But that is not even the case, as this modeling approach has already been applied in growth experiments (Tonner et al., 2020). This is not properly stated in the discussion (L445), where the “omnibus” approach is presented as something entirely new and different, without referring to any similar studies from outside the research group of the authors.
## The introduction does not properly present the novelty of the research
The introduction is mostly focused on justifying the relevance of Listeria monocytogenes and why growth models for this microorganism at refrigeration temperatures are important. However, this does not have high novelty. For instance, ComBase has more than 5,000 growth curves of L. monocytogenes between 0 and 14ºC.
The novelty could be attributed to the modeling approach. However, the authors do not properly review previous studies comparing their methodology against others. The comparison is limited to a couple of sentences (L68-71), and does not include any work from outside their research group.
Minor comments
---------------------
- L29-30: I disagree with this claim. In my opinion, this is actually one disadvantage of global approaches. In a two-step approach one gets “intermediate” parameters and has to check they follow the model equations (e.g. the secondary model). This is not the case in global approaches, where the algorithm provides global parameter estimates. It is true these “intermediate” parameters can be calculated a posteriori, or deviations can be identified based on residual analysis. But these calculations are an additional calculation, making identifying these deviations more complex than in two-step approaches where this check is somewhat implemented in the model.
- L59: Please define “primary growth model”.
- L79: Please define what does “variability” means in this context and why it is relevant for risk assessment.
- L80-81: If there is evidence that variability is higher at lower temperatures, why doesn’t the model implement this hypothesis? Instead, it models variance as independent from temperature. If the model based on this hypothesis correctly describes the data, isn’t it refuting this hypothesis?
- L81: Please be more specific with what “low temperatures” means.
- L108: Please be more specific with what a “biological replicate” is in this context. Do they all come from the same colony or from different ones? This can be of high relevance for the interpretation of variability.
- L111: The symbols in the equations (model parameters, variables…) are not defined in the text.
- L118: Please provide an additional description of the model. Why is there a number 4 or a number 25 in the model?
- L135: The version of RStudio is irrelevant, as it is just an IDE for the R programming language. Please provide the version of R instead.
- L141: Why are the authors using a different reference (17) than above (L114; 8, 15)? Isn’t it the same model? If it is not, the new equation should be included, and properly justified why a different primary model is used in both approaches.
- L141-142: Please describe in detail the method to “define and assess for goodness of fit” the secondary models. Especially for the lag phase duration
- L149: Why does the initial count depend on temperature (u_j)? If the pre-incubation conditions are the same for every experiment, shouldn’t they result in the same number of cells?
- L150: What is the reasoning behind the secondary model for the lag phase? There is plenty of evidence proving that the pre-incubation conditions affect the lag phase duration (e.g. (Walker et al., 1990)). However, the secondary model for the lag phase only considers the “instantaneous” temperature, without any hypothesis regarding the previous history of the cells. How does that affect the model? Is it only valid for cells pre-incubated at the same conditions as the one in this experiment? How useful is the model for shelf-life estimation, then?
- L190: I don’t think “separately” the best word here. Especially considering this model fits a model to the 5 strains at the same time, whereas the previous one fits five models separately for each strain.
- L195-196: Why does strain variability only affect the intercept of the secondary model? Isn’t it reasonable that strain variability will only affect cell sensitivity to temperature changes?
- L209-211: Is this reasonable? The initial count depends on the pre-incubation conditions. If the experimental protocol is the same for every strain, why would it result in different initial counts?
- L211-212: Is this a reasonable hypothesis? Why would the shift in Nmax would be related to the shift in lambda?
- L233-235: Please provide further information here. Version of R, version of nlme (and reference to the package), method used for model fitting (REML or ML), any possible controls that were defined…
- L241: Please provide the same information for the other strains tested.
- L247-248: How is this possible? How could the authors estimate Nmax with reasonable standard errors using just a primary model without observing it?
- Figure 1: There is a mismatch between the plot and the legend in the symbol for 7.8ºC.
- L257: Is the “Huang growth rate parameter” different from the usual “growth rate” used in predictive microbiology?
- L264-266: Doesn’t this contradict the authors hypothesis to build the model that strain variability affects the intercept of the secondary model for \mu but not its slope (L195-196)?
- L266-268: Doesn’t this contradict the conclusions gathered from the literature review presented in the introduction, which stated that “There is also evidence that variability is higher at low temperatures” (L80). Does a sqrt-transformation get rid of this heteroskedasticity?
- Table 1: I think there is a typo in the units (it should be ln(CFU)/h, not ln(CFU/h)).
- Table 1: Is giving a standard error up to the 7th decimal point for \mu realistic?
- Table 1: How was the standard error for \mu calculated?
- Figure 2: Why is there a shift in the x-direction? Is there a jitter in the data points?
- Tables 2-6: Please include units for every model parameter.
- L343-344: Would it be a valid hypothesis that Y0 is defined by the pre-incubation conditions, that are quite controlled, whereas Ymax is the outcome of the growth experiment where variability and uncertainty may be more relevant?
- L360-361: This is against the scientific literature and should be discussed at length.
- L410: Doing this analysis in the original scale is misleading. It is generally accepted in the field that a sqrt-transformation is required to stabilize the variance in the growth rate.
- L413: Is the fact it has a “shiny” interface relevant? Or just the parameter estimates?
- L413: Did that study use the same strains or different ones?
References
-----------------
Aryani, D. C., den Besten, H. M. W., Hazeleger, W. C., & Zwietering, M. H. (2015). Quantifying strain variability in modeling growth of Listeria monocytogenes. International Journal of Food Microbiology, 208, 19–29. https://doi.org/10.1016/j.ijfoodmicro.2015.05.006
Garre, A., Zwietering, M. H., & den Besten, H. M. W. (2020). Multilevel modelling as a tool to include variability and uncertainty in quantitative microbiology and risk assessment. Thermal inactivation of Listeria monocytogenes as proof of concept. Food Research International, 137, 109374. https://doi.org/10.1016/j.foodres.2020.109374
Jaloustre, S., Guillier, L., Morelli, E., Noël, V., & Delignette-Muller, M. L. (2012). Modeling of Clostridium perfringens vegetative cell inactivation in beef-in-sauce products: A meta-analysis using mixed linear models. International Journal of Food Microbiology, 154(1), 44–51. https://doi.org/10.1016/j.ijfoodmicro.2011.12.013
McElreath, R. (2016). Statistical rethinking: A Bayesian course with examples in R and Stan. CRC Press/Taylor & Francis Group.
Nunes Silva, B., Cadavez, V., Teixeira, J. A., & Gonzales-Barron, U. (2020). Effects of Essential Oils on Escherichia coli Inactivation in Cheese as Described by Meta-Regression Modelling. Foods, 9(6), 716. https://doi.org/10.3390/foods9060716
Tonner, P. D., Darnell, C. L., Bushell, F. M. L., Lund, P. A., Schmid, A. K., & Schmidler, S. C. (2020). A Bayesian non-parametric mixed-effects model of microbial growth curves. PLOS Computational Biology, 16(10), e1008366. https://doi.org/10.1371/journal.pcbi.1008366
van Boekel, M. A. J. S. (2020). On the pros and cons of Bayesian kinetic modeling in food science. Trends in Food Science & Technology, 99, 181–193. https://doi.org/10.1016/j.tifs.2020.02.027
Walker, S. J., Archer, P., & Banks, J. G. (1990). Growth of Listeria monocytogenes at refrigeration temperatures. The Journal of Applied Bacteriology, 68(2), 157–162. https://doi.org/10.1111/j.1365-2672.1990.tb02561.x
Author Response
In the manuscript entitled “Omnibus modelling of Listeria monocytogenes growth rates at low temperatures”, the authors apply a global approach to describe the growth kinetics of L. monocytogenes are refrigeration temperatures. Although the approach could be of interest as it enables to include biological variability, there are severe deficiencies in experimental design and model validation in this study. Moreover, some hypotheses in model building should be better reasoned, and there
Response: We have carefully considered all the issues this reviewer has raised and hopefully we have answered them appropriately.
(note the apparent curtailment of the final sentence above was present in the version of the review visible to us as authors)
Major comments
## The number of strains chosen in the study is too small to properly describe between-strain variability
One of the advantages of multilevel models is the explicit definition of variance components to describe the variability in the microbial response. The authors make a link in several parts of the manuscript between their modelling approach and the study of variability (e.g. L79 or L209-211). It is important to highlight that this is an empirical model, whose quality is (at most) as good as the quality of the data used to build it. For instance, when describing a growth curve, an article would never be accepted if it had just a couple of time points per experiment. The same logic applies to studies on variability: their estimates can only be precise if the set of strains included in the study are representative of the actual population of Listeria strains.
Response: This study was never intended to be an exhaustive study of between-strain variability in Listeria monocytogenes. The abstract and study objective makes it clear that only five strains were selected. The objective was to compare omnibus modelling to a conventional sequential first order / second order modelling for growth studies to investigate what, if any, additional insight can be gained from omnibus modelling applied to an experimental data set. Nowhere in the manuscript was it stated that the objective of the study was to estimate between-strain variability in L. monocytogenes. This would be a very ambitious objective that would require the combination of data coming from multiple strains.
I believe this is the reason the authors conclude there is no between-strain variability in growth (L360-361). This is a surprising result that goes against most of the scientific literature, which has observed differences in the growth characteristic at the between-strain level. This is most likely a result of the narrow experimental design (just 5 strains , whereas e.g. (Aryani et al., 2015) studied 20 strains). Therefore, the estimates of between-strain variability would be an underestimation. Considering that variability can be as relevant as kinetic parameters for shelf life estimation and risk assessment (specially for a microorganism with the ecology of L.monocytogenes), this model could not be used as a overarching model that generally describes the response of Listeria monocytogenes. Then, it could be used as just one model to describe the response of a single strain. But there are already thousands of growth curves published for L.monocytogenes at this temperature range.
Response: Again, this study was never intended to be an exhaustive study of between-strain variability in Listeria monocytogenes. We never implied or stated in the manuscript that we were proposing an “overarching model for growth of L. monocytogenes”. If this were our objective, we would have chosen an alternative approach such as the meta-analysis path. Our objective was to compare omnibus modelling to a conventional sequential first order / second order modelling for growth studies to investigate what, if any, additional insight can be gained from omnibus modelling applied to an experimental data set.
## The selection of the method for model validation is surprising
In empirical models, model validation is done to assess whether the model is able to predict the microbial response at conditions different to the ones used to build the model (i.e. model fitting). For instance, a model fitted to data at 5 and 10ºC, can be validated using independent data at 7ºC to assess whether the model is valid for temperatures within 5 and 10ºC.
I find it surprising how validation is interpreted by the authors. They used as training set 2 of the replicates, using the third one as validation (L226-229). That is, they are using exactly the same conditions (storage time, storage temperature, strain, pre-incubation conditions…) for training and validation. From my understanding, this serves to validate that the model would be good at predicting a fourth replication of this experiment, nothing more. Then, how useful is this model for risk assessment?
Response: The authors are well aware that model validation must be performed using levels of environmental conditions different from those that generated the model. However, at a first instance, we did not separate the 7ºC data from the full data set because that would have left us with only two temperature points to fit the model (which would affect the tested variability between the five strains). So, we opted instead for separating the third replicate, knowing well that it is not the conventional way. Nonetheless, as suggested by the reviewer, we have now performed a second validation study, by fitting the omnibus model with the 4ºC and 12ºC data, and validating it using the entire 7ºC data. Despite being anchored by only two temperature points, the model validated very well, and this gives us assurance that the omnibus model architecture is satisfactory. See the new Figure 3 and an updated description of the validation model from line 236 onwards.
## There are several hypotheses in model construction that are arguable
(listed in the minor comments below)
## The use of the term “omnibus” gives a misleading exaggeration of novelty
The authors use the term “omnibus” to define their approach, a term that is not broadly used in the field. This gives the impression that this approach is entirely novel. To my knowledge, it has only been used in the two articles referred by the authors (L74), which were written by the same authors.
Response: For at least 10 years, the authors have been aware of the use of multilevel modelling for jointly fitting a primary and a secondary model. So, at least for us, this approach is not new at all (as can be attested in several Gonzales-Barron et al publications since 2013). Nonetheless, it is still not widely used nor has it been sufficiently advocated. The term "omnibus" modelling referring to multilevel or mixed modelling in predictive microbiology, was first used by Juneja et al. (2003) (USDA research group) and not by the authors of this paper. From 2013, Gonzales-Barron and co-workers continued to use the term in subsequent papers such as Juneja et al. (2013; 2015; 2016) and others. Later on, Huang, L. (2015) (another USDA researcher) used the term "one-step" or "global" direct approach to refer to the same modelling strategy (i.e., the determination of kinetic parameters using all experimental data). These new terms (one-step and global) somehow added to the confusion. Thus, we do not believe that omnibus modelling is a new approach, neither is it our intention to make it look “exaggeratedly novel”. Multilevel modelling (now using the broader term) is by no means a new approach in predictive microbiology (as suggested in Garre et al., 2020), but it has been instead long ago known as a suitable method for extracting microbial kinetic parameters - despite it has not been widely applied.
The terms "multilevel" or "hierarchical" are purely statistical terms of a general mixed model that can be applied to any scientific field. Therefore, thes authors prefer to stick to the older term "omnibus" which is more specific and familiar to predictive microbiologists. The authors believe there is no need to introduce newer terms to add to the confusion, when another term has been already in use.
Relevant References:
Juneja, V. K., Marks, H. M. Marks and Mohr,T. 2003. Predictive Thermal Inactivation Model for Effects of Temperature, Sodium Lactate, NaCl, and Sodium Pyrophosphate on Salmonella Serotypes in Ground Beef. Applied and Environmental Microbiology, Volume 69, Issue 9, Pages 5138-5156
Juneja, V., Cadavez, V., Gonzales-Barron, U., and Mukhopadhyaya, S. (2015). Effect of pH, sodium chloride and sodium pyrophosphate on the thermal resistance of Escherichia coli O157:H7 in ground beef. Food Research International 69, 289-304.
Juneja, V., Gonzales-Barron, U., Butler, F., Yadav, A., and Friedman, M. (2013). Predictive thermal inactivation model for the combined effect of temperature, cinnamaldehyde and carvacrol on starvation-stressed multiple Salmonella serotypes in ground chicken. International Journal of Food Microbiology, 165(2): 184-199.
Huang L. 2015. Direct construction of predictive models for describing growth of Salmonella
enteritidis in liquid eggs—a one-step approach. Food Control 57:76–81.
However, the application of mixed effects models in predictive microbiology is not new. For instance, most meta-regression studies have applied this approach (e.g. (Jaloustre et al., 2012; Nunes Silva et al., 2020). It could be argued this method is more “advanced” than those, as it includes the errors at different levels. However, the methodology applied by the authors is similar to the one applied in several recent studies (Garre et al., 2020; van Boekel, 2020). It might be argued that the novelty comes from this methodology being adapted for growth, whereas previous studies applied it for inactivation (although this argument is quite weak in my opinion). But that is not even the case, as this modeling approach has already been applied in growth experiments (Tonner et al., 2020). This is not properly stated in the discussion (L445), where the “omnibus” approach is presented as something entirely new and different, without referring to any similar studies from outside the research group of the authors.
Response: As discussed above, the authors have used omnibus modelling in predictive microbiology since 2013, so it is not a new discovery for us, nor do we want it to appear it looks as new. Thus, the approach we applied does not come from the 2020 papers the reviewer refers to. Nonetheless, we have broadened to include some of the references highlighted starting line 459
## The introduction does not properly present the novelty of the research
The introduction is mostly focused on justifying the relevance of Listeria monocytogenes and why growth models for this microorganism at refrigeration temperatures are important. However, this does not have high novelty. For instance, ComBase has more than 5,000 growth curves of L. monocytogenes between 0 and 14ºC.
Response We recognise that there is much growth data for L. monocytogenes in the literature. However it is striking that the recent guidance documents on how to conduct laboratory shelf-life studies on L. monocytogenes continue to describe what are essentially traditional sequential first order and second order modelling approaches. The objective of this paper is to investigate what utility omnibus modelling has to complement conventional sequential first order / second order modelling for growth / challenge studies.
The novelty could be attributed to the modeling approach. However, the authors do not properly review previous studies comparing their methodology against others. The comparison is limited to a couple of sentences (L68-71), and does not include any work from outside their research group.
Response: As above, there is now an expanded section in the discussion section starting line 459 comparing our work to related studies. Nonetheless, bearing in mind that USDA researchers and Gonzales-Barron and co-workers have been pioneers in omnibus modelling for predictive microbiology, it is understandable that their works be often cited.
Minor comments
---------------------
L29-30: I disagree with this claim. In my opinion, this is actually one disadvantage of global approaches. In a two-step approach one gets “intermediate” parameters and has to check they follow the model equations (e.g. the secondary model). This is not the case in global approaches, where the algorithm provides global parameter estimates. It is true these “intermediate” parameters can be calculated a posteriori, or deviations can be identified based on residual analysis. But these calculations are an additional calculation, making identifying these deviations more complex than in two-step approaches where this check is somewhat implemented in the model.
Response: We disagree with the point made by the reviewer. Omnibus modelling does not eliminate the need for following the sequence primary – secondary modelling. We strongly advocate its continued use. Still, even when extracting the kinetic parameters by omnibus modelling (mixed effects), it is necessary to evaluate the “best” primary and secondary models to represent the data. So anyway we need to estimate the intermediate parameters and appraise their relationship with the environmental conditions. Omnibus modelling is not done blindly (at least not by this group of researchers). With omnibus modelling, the errors associated with the kinetic parameters (say, mu_max at different temperatures) are not lost, as they are in the two-step approach. In addition, the modeller can take into account both the data and error structure.
L59: Please define “primary growth model”.
Response: A clarification of what is meant by a primary growth model has been added to L61
L79: Please define what does “variability” means in this context and why it is relevant for risk assessment.
Response: variability is used here in its normal statistical sense. We are not sure of the comment relating to risk assessment – we are not at that section of the paper referring to risk assessment.
L80-81: If there is evidence that variability is higher at lower temperatures, why doesn’t the model implement this hypothesis? Instead, it models variance as independent from temperature. If the model based on this hypothesis correctly describes the data, isn’t it refuting this hypothesis?
Response: Random effects associated with temperature was tested but variance was non significant so it was dropped from the model.
L81: Please be more specific with what “low temperatures” means.
Response: The specific temperatures referred to have been added to the text
L108: Please be more specific with what a “biological replicate” is in this context. Do they all come from the same colony or from different ones? This can be of high relevance for the interpretation of variability.
Response: For a biological replicate, the researcher went back to the original isolate stored at -20 Deg C and generated new starting inoculum.
L111: The symbols in the equations (model parameters, variables…) are not defined in the text.
Response: The meanings of the parameters “Nt”, “N0”, “Nmax”, “lag”, and “μmax” have been added to the text (line 119)
L118: Please provide an additional description of the model. Why is there a number 4 or a number 25 in the model?
Response: These values have been replaced by a constant, α, which can have a value of either 4 or 25 in line with as described by Huang in reference number 15. An explanation has been added at lines 122 and 134 and equations 2 and 3 modified accordingly.
L135: The version of RStudio is irrelevant, as it is just an IDE for the R programming language. Please provide the version of R instead.
Response: The version of R has been added where relevant and a cite to the R Core Group (2013) as well.
L141: Why are the authors using a different reference (17) than above (L114; 8, 15)? Isn’t it the same model? If it is not, the new equation should be included, and properly justified why a different primary model is used in both approaches.
Response: The same primary model has been applied, the references have been aligned.
L141-142: Please describe in detail the method to “define and assess for goodness of fit” the secondary models. Especially for the lag phase duration
Response: The text has been amended to “simple data-driven first and second order polynomial models”. As there was only three temperatures, the models were simply fitting a model (linear in the case of growth rate) to three data points.
L149: Why does the initial count depend on temperature (u_j)? If the pre-incubation conditions are the same for every experiment, shouldn’t they result in the same number of cells?
Response: The initial count does not depend on temperature. What we are saying is that logN0 randomly varies condition to condition (and, in this study, a condition is defined only by the temperature, but the condition is a class variable, and it is not the temperature itself). In practical terms, the random effects put on logN0 (with condition as clustering variable) are extracting the random shifts in inoculum size occurred in the experimental runs, since in every biological replicate, a new inoculum was prepared.
L150: What is the reasoning behind the secondary model for the lag phase? There is plenty of evidence proving that the pre-incubation conditions affect the lag phase duration (e.g. (Walker et al., 1990)). However, the secondary model for the lag phase only considers the “instantaneous” temperature, without any hypothesis regarding the previous history of the cells. How does that affect the model? Is it only valid for cells pre-incubated at the same conditions as the one in this experiment? How useful is the model for shelf-life estimation, then?
Response: As above, as there was only three temperatures, the models were simply fitting a model (linear in the case of growth rate) to three data points. If we were to address the issue raised by the reviewer, it would require a completely different experimental design. As stated previously, we are not attempting to build a comprehensive model for the growth of L. monocytogenes across a whole range of factors. The objective was to compare omnibus modelling to a conventional sequential first order / second order modelling for growth studies to investigate what, if any, additional insight can be gained from omnibus modelling applied to an experimental data set
L190: I don’t think “separately” the best word here. Especially considering this model fits a model to the 5 strains at the same time, whereas the previous one fits five models separately for each strain.
Response: Separately was used here referring to the five strains – its place in the sentence has been changed to reflect that (line 200)
L195-196: Why does strain variability only affect the intercept of the secondary model? Isn’t it reasonable that strain variability will only affect cell sensitivity to temperature changes?
Response: The variance of random effects put to temperature was not significant; meaning that strains did not produce slopes that were statistically different. This is why this candidate model was disregarded and not presented in the manuscript.
L209-211: Is this reasonable? The initial count depends on the pre-incubation conditions. If the experimental protocol is the same for every strain, why would it result in different initial counts?
Response: Yes, it is reasonable. In every run, the same protocols for activation/pre-incubation conditions were used. Nonetheless, there is always some experimental error occurring, and this is precisely one of the good things of omnibus modelling. This noise affecting log N0 can be removed.
L211-212: Is this a reasonable hypothesis? Why would the shift in Nmax would be related to the shift in lambda?
Response: This was not a hypothesis made by the authors. The authors chose a natural (flexible) covariance structures, and this correlation was observed from the data.
L233-235: Please provide further information here. Version of R, version of nlme (and reference to the package), method used for model fitting (REML or ML), any possible controls that were defined…
Response: This has been updated in the text line 145, 245
L241: Please provide the same information for the other strains tested.
Response: Two plots are now shown. We are of the view that if all five plots are shown it would replicate too much what is shown in the subsequent plot.
L247-248: How is this possible? How could the authors estimate Nmax with reasonable standard errors using just a primary model without observing it?
Response: Generally there was sufficient curvature in the data approaching the stationary phase to allow Nmax to be estimated by the Huang model. In a small number of cases this was not possible and the reduced form of the primary model used to estimate growth rate.
Figure 1: There is a mismatch between the plot and the legend in the symbol for 7.8ºC.
Response: Replaced the asterisk in the figure caption with the asterisk as used in the plot
L257: Is the “Huang growth rate parameter” different from the usual “growth rate” used in predictive microbiology?
Response: No. We have removed the name of the model to avoid confusion.
L264-266: Doesn’t this contradict the authors hypothesis to build the model that strain variability affects the intercept of the secondary model for \mu but not its slope (L195-196)?
Response: We used the results of the first order modelling to guide the development of the omnibus modelling. However, the amount of data available did not allow to have random effects on both intercept and slope. When random effects were placed only on the temperature slope, the variance was not significant.
L266-268: Doesn’t this contradict the conclusions gathered from the literature review presented in the introduction, which stated that “There is also evidence that variability is higher at low temperatures” (L80). Does a sqrt-transformation get rid of this heteroskedasticity?
Response: If more strains were used in the study, perhaps we would have seen this effect. Even with the transformation, heteroskedasticity will not be fully removed.
Table 1: I think there is a typo in the units (it should be ln(CFU)/h, not ln(CFU/h)).
Response: Changed to ln(CFU)/h
Table 1: Is giving a standard error up to the 7th decimal point for \mu realistic?
Response: Excluding the leading zeroes, most of the numbers are given to three significant digits only. At least two significant digits seems necessary, and three does not seem excessive. It was also intended to show that standard errors were virtually zero, and therefore the clustering variable did not significantly affect the given parameter.
Table 1: How was the standard error for \mu calculated?
Response: There were three estimates for \mu, one for each replicate. The standard deviation is just the standard deviation of these three estimates. This has been clarified in the table caption. (Standard errors for each individual estimate were calculated when the model was fit in R using the non-linear regression function nls(), but not included in the table).
Figure 2: Why is there a shift in the x-direction? Is there a jitter in the data points?
Response: Yes a slight shift was introduced to improve readability – this is now referred to in the caption
Tables 2-6: Please include units for every model parameter.
Response: Done throughout the paper
L343-344: Would it be a valid hypothesis that Y0 is defined by the pre-incubation conditions, that are quite controlled, whereas Ymax is the outcome of the growth experiment where variability and uncertainty may be more relevant?
Response: with replication, Yo is not fully defined by the pre-incubation conditions, and for Ymax, we are confident that the higher variability observed in the maximum concentration may be linked to the absence of an observed stationary phase in the 7.8 ºC data.
L360-361: This is against the scientific literature and should be discussed at length.
Response: What the results say is that for these five strains, there was not much variability between strains, we did not set out to compare an exhaustive strain bank.
L410: Doing this analysis in the original scale is misleading. It is generally accepted in the field that a sqrt-transformation is required to stabilize the variance in the growth rate.
Response: The ANOVA was done separately at each temperature on the growth rate data and we tested for equality of variances using a Levene test (line 145) and normality of residuals. We considered doing a two way ANOVA and considered transforming the data for that but in fact the two way was not necessary given the obvious differences at the three temperatures.
L413: Is the fact it has a “shiny” interface relevant? Or just the parameter estimates?
Response: We are simply reporting that a relevant shiny is available. The shiny interface is relevant for the interested readers who may want to find the reported estimates by themselves.
L413: Did that study use the same strains or different ones?
Response: The current study used three EURL strains. The SHINY app used data from multiple strains
References
-----------------
Aryani, D. C., den Besten, H. M. W., Hazeleger, W. C., & Zwietering, M. H. (2015). Quantifying strain variability in modeling growth of Listeria monocytogenes. International Journal of Food Microbiology, 208, 19–29. https://doi.org/10.1016/j.ijfoodmicro.2015.05.006
Garre, A., Zwietering, M. H., & den Besten, H. M. W. (2020). Multilevel modelling as a tool to include variability and uncertainty in quantitative microbiology and risk assessment. Thermal inactivation of Listeria monocytogenes as proof of concept. Food Research International, 137, 109374. https://doi.org/10.1016/j.foodres.2020.109374
Jaloustre, S., Guillier, L., Morelli, E., Noël, V., & Delignette-Muller, M. L. (2012). Modeling of Clostridium perfringens vegetative cell inactivation in beef-in-sauce products: A meta-analysis using mixed linear models. International Journal of Food Microbiology, 154(1), 44–51. https://doi.org/10.1016/j.ijfoodmicro.2011.12.013
McElreath, R. (2016). Statistical rethinking: A Bayesian course with examples in R and Stan. CRC Press/Taylor & Francis Group.
Nunes Silva, B., Cadavez, V., Teixeira, J. A., & Gonzales-Barron, U. (2020). Effects of Essential Oils on Escherichia coli Inactivation in Cheese as Described by Meta-Regression Modelling. Foods, 9(6), 716. https://doi.org/10.3390/foods9060716
Tonner, P. D., Darnell, C. L., Bushell, F. M. L., Lund, P. A., Schmid, A. K., & Schmidler, S. C. (2020). A Bayesian non-parametric mixed-effects model of microbial growth curves. PLOS Computational Biology, 16(10), e1008366. https://doi.org/10.1371/journal.pcbi.1008366
van Boekel, M. A. J. S. (2020). On the pros and cons of Bayesian kinetic modeling in food science. Trends in Food Science & Technology, 99, 181–193. https://doi.org/10.1016/j.tifs.2020.02.027
Walker, S. J., Archer, P., & Banks, J. G. (1990). Growth of Listeria monocytogenes at refrigeration temperatures. The Journal of Applied Bacteriology, 68(2), 157–162. https://doi.org/10.1111/j.1365-2672.1990.tb02561.x
Reviewer 4 Report
The present work reports on an one-step modelling approach for the quantitative description of the growth of Listeria monocytogenes as a function of temperatures, with low (i.e. refrigeration or abusive) temperatures being taken into account, while potential strain-dependent effects also were attempted to be assessed.
Overall, the approach is interesting as an alternative (to the conventional two-step approach) modelling approach, since it allows for minimal (if any) loss of information while at the same time potential systematic errors associated with the environmental conditions can also be identified (and properly treated). Nonetheless, according to this reviewer's view, there are some important limitations of this study, not allowing for general conclusions to be drawn (as practiced herein).
Major concerns:
- It is not clear why these specific five strains were selected in this study...The authors mention in L88 that "The strains were chosen to be representative of ongoing challenge studies...". Where? Specifically in Ireland? And why is this a sound criterion for strain selection. Furthermore, in L436-438 the authors also mention that two of the five strains (almost 50%) were previously characterized as robust in terms of their growth ability at low temperatures. This demonstrates a bias in strain selection which (i) does not really allow for the objective assessment of strain variability in growth rates (to which a lot of reference has been made in this manuscript), and (ii) somehow explains the really unexpected (and in contrast to what reported in the literature) finding of this study, according to which strain variability was higher with increasing storage temperature. Anyway, and beyond the above, the number of the studied strains was very low to justify an accurate strain variability assessment, and thus, the extensive commentary made in this work regarding this issue.
- This is exclusively a culture broth study, and no discussion is provided regarding the necessity of validating the finding in real (or model) food systems. The fact that this study involved a limited number of strains and culture in broth, renders its findings only preliminary in nature.
- The authors refer to μmax as exponential growth rate or simply growth rate (whereas the accurate term is maximum specific growth rate), and throughout the manuscript (with the exception of Table 8), the associated unit is incorrectly identified (it should be h-1 and not ln CFU h-1) .
- L231, 388: No information is provided (neither a pertinent reference) on how the performance indices of Af and Bf were determined. According to Ross (1996), these metrics should be estimated based on the generation times (as derived from the μmax values), but the reviewer cannot be sure that this was the case here.
- In the abstract the authors mention that "There were significant differences in growth rate depending on the strain at all temperatures" but in the main body of the manuscript, a different (contradictory) statement is made, namely: "There was very little difference in growth rates between strains at 4.5 °C)"...
Minor comments
- L38: it should be revised to "facultatively anaerobic"
- L46: this is where the acronym "RTE" should appear for the first time (and not in L51)
- L55-56: there are additional guidance documents with regard to the design of food safety challenge studies involving L. monocytogenes, e.g., Scott et al., 2005 (Food Protection Trends 25, 818-825); NACMCF, 2010 (Journal of Food Protection 73, 140-202).
- General comment: a model is fitted to data and not the opposite
- General comment: "between" is used for comparisons made for two items while "among" for >2 items; thus, the appropriate phrase throughout the manuscript is "among strains" and/or "among temperatures/environmental conditions"
- L81: there are more studies on the strain variability in the growth behavior of L. monocytogenes under different environmental conditions, e.g., Lianou et al., 2006 (Journal of Food Protection 69, 2640-2647); Aryani et al., 2015 (International Journal of Food Microbiology 208, 19-29)
- L98: why did you choose to prepare the cultures for inoculum preparation starting from colonies picked up from a selective medium? This is strongly discouraged, and there was actually no need to do so here, since you start from pure strain cultures.
- Not clear why these exact growth temperatures were selected for modelling, namely 4.5, 7.8 and 12°C. Was this the initial goal or the initial objective was to study 4, 8 and 12°C, and the aforementioned values were the actually recorded ones? This should be clarified.
- L106 and 109: by "spotted" you mean "inoculated and surface plated"?
- You should define all the parameters included in equations 1 and 2
- L185: correct to "goodness-of-fit"
- L210 (and wherever else applicable): use "inter-strain" instead of "between-strain"
- L282: no need to report the exact P-value but a statement of "P<0.05" would be sufficient
- Table 1: no need to report so many decimal digits for the standard deviation in parentheses (four decimal digits, as in the average value would be sufficient)
Author Response
The present work reports on an one-step modelling approach for the quantitative description of the growth of Listeria monocytogenes as a function of temperatures, with low (i.e. refrigeration or abusive) temperatures being taken into account, while potential strain-dependent effects also were attempted to be assessed.
Overall, the approach is interesting as an alternative (to the conventional two-step approach) modelling approach, since it allows for minimal (if any) loss of information while at the same time potential systematic errors associated with the environmental conditions can also be identified (and properly treated). Nonetheless, according to this reviewer's view, there are some important limitations of this study, not allowing for general conclusions to be drawn (as practiced herein).
Response: Thank you for this favourable review of the approach we present – this is the main focus of the paper. We hope we have answered your reservations listed below to your satisfaction.
Major concerns:
It is not clear why these specific five strains were selected in this study...The authors mention in L88 that "The strains were chosen to be representative of ongoing challenge studies...". Where? Specifically in Ireland? And why is this a sound criterion for strain selection. Furthermore, in L436-438 the authors also mention that two of the five strains (almost 50%) were previously characterized as robust in terms of their growth ability at low temperatures. This demonstrates a bias in strain selection which (i) does not really allow for the objective assessment of strain variability in growth rates (to which a lot of reference has been made in this manuscript), and (ii) somehow explains the really unexpected (and in contrast to what reported in the literature) finding of this study, according to which strain variability was higher with increasing storage temperature. Anyway, and beyond the above, the number of the studied strains was very low to justify an accurate strain variability assessment, and thus, the extensive commentary made in this work regarding this issue.
Response: As discussed extensively with the other reviewers, the objective of this paper was to investigate what utility omnibus modelling has to complement conventional sequential first order / second order modelling for growth / challenge studies. The data set analysed in this paper presented an ideal opportunity to do just that. We never set out to conduct an extensive strain variability assessment but to demonstrate that the modelling approach can explore such issues.
This is exclusively a culture broth study, and no discussion is provided regarding the necessity of validating the finding in real (or model) food systems. The fact that this study involved a limited number of strains and culture in broth, renders its findings only preliminary in nature.
Response: As above, We never set out to conduct an extensive strain variability or food matrix assessment but to demonstrate that the modelling approach can explore such issues.
The authors refer to μmax as exponential growth rate or simply growth rate (whereas the accurate term is maximum specific growth rate), and throughout the manuscript (with the exception of Table 8), the associated unit is incorrectly identified (it should be h-1 and not ln CFU h-1) .
Response: This has been standardised and changed throughout the text.
L231, 388: No information is provided (neither a pertinent reference) on how the performance indices of Af and Bf were determined. According to Ross (1996), these metrics should be estimated based on the generation times (as derived from the μmax values), but the reviewer cannot be sure that this was the case here.
Response: This reference has been added to explain the Af and Bf calculation, However, the only place we use them is a footnote to figure 3
In the abstract the authors mention that "There were significant differences in growth rate depending on the strain at all temperatures" but in the main body of the manuscript, a different (contradictory) statement is made, namely: "There was very little difference in growth rates between strains at 4.5 °C)"...
Response: ANOVA of the growth rates estimated by the did show small significant differences whereas the omnibus modelling indicated smaller variation – this difference in outcomes depending on the modelling approach is discussed in the discussion. We have amended the abstract to reflect this.
Minor comments
L38: it should be revised to "facultatively anaerobic"
Response: Agreed and revised to “facultatively anaerobic”.
L46: this is where the acronym "RTE" should appear for the first time (and not in L51)
Response: Acronym now defined at first appearance of “ready to eat”.
L55-56: there are additional guidance documents with regard to the design of food safety challenge studies involving L. monocytogenes, e.g., Scott et al., 2005 (Food Protection Trends 25, 818-825); NACMCF, 2010 (Journal of Food Protection 73, 140-202).
Response: Thank you, we have included them in the introduction
General comment: a model is fitted to data and not the opposite
Response: The language in the last paragraph of the introduction has been amended to avoid this confusion.
General comment: "between" is used for comparisons made for two items while "among" for >2 items; thus, the appropriate phrase throughout the manuscript is "among strains" and/or "among temperatures/environmental conditions"
Response: Some instances of “between” have been changed to “among” where this distinction can be drawn. In other cases pairwise comparisons are being made between all possible pairs of strains and the language has been left as it stands. Finally, as noted in a later response by reviewer 4, some instances of “between strain” have been amended to “inter-strain” where appropriate.
L81: there are more studies on the strain variability in the growth behavior of L. monocytogenes under different environmental conditions, e.g., Lianou et al., 2006 (Journal of Food Protection 69, 2640-2647); Aryani et al., 2015 (International Journal of Food Microbiology 208, 19-29)
Response: Thank you, we have added these
L98: why did you choose to prepare the cultures for inoculum preparation starting from colonies picked up from a selective medium? This is strongly discouraged, and there was actually no need to do so here, since you start from pure strain cultures.
Response: We agree with the reviewer that there was no need to culture on the selective agar but that the important point is that the cells that went into the experiment not straight from the selective media but were grown first on non-selective BHI
Not clear why these exact growth temperatures were selected for modelling, namely 4.5, 7.8 and 12°C. Was this the initial goal or the initial objective was to study 4, 8 and 12°C, and the aforementioned values were the actually recorded ones? This should be clarified.
Response: Yes the nominal temperatures were 4 8 and 12 but we included data loggers in the incubator and opted to report the mean temperature recorded
L106 and 109: by "spotted" you mean "inoculated and surface plated"?
Response: pipetted – amended in text
You should define all the parameters included in equations 1 and 2
Response: As noted with reviewer 3, definitions for the parameters have now been added to the text.
L185: correct to "goodness-of-fit"
Response: “goodness of fit” now reads “goodness-of-fit”.
L210 (and wherever else applicable): use "inter-strain" instead of "between-strain"
Response: This has been changed in the two places it was used.
L282: no need to report the exact P-value but a statement of "P<0.05" would be sufficient
Response: The captions for Table 1 and Figure 2 have been amended to state the significance threshold used, as suggested.
Table 1: no need to report so many decimal digits for the standard deviation in parentheses (four decimal digits, as in the average value would be sufficient)
Response: As with reviewer three Excluding the leading zeroes, most of the numbers are given to three significant digits only. At least two significant digits seems necessary, and three does not seem excessive. It was also intended to show that standard errors were virtually zero, and therefore the clustering variable did not significantly affect the given parameter.
Round 2
Reviewer 3 Report
The authors have done limited changes in the manuscript. The major change with respect to the original version of the manuscript is the use of a different training set (two temperatures instead of two replicates). However, the authors report exactly the same parameter estimates (e.g. Table 2). This is simply impossible, and I expect this error is due to the authors forgetting to update these tables.
Besides that, the authors have made minor adjustments and provided a lengthy rebuttal letter. In my opinion, this does not justify the lack of contextualization and novelty of the work. The “omnibus” is currently applied by most research groups specialized in predictive microbiology, and plenty of research articles have already made the comparison that is presented as the main goal in this article reaching the same conclusions (in many cases, using a dataset that allowed a better comparison). The main difference is that here the authors use the term “omnibus”, whereas other researchers use “one-step”, “global” or similar terms.
Major comments
----------------------
## It is impossible that a change in the training set does not affect the parameter estimates
In the original version of the manuscript, the authors used two replicates for model fitting and one for model validation. Based on my comments, they claim to have modified this method, using two temperatures for model fitting and one for model validation (L257-261). However, they report exactly the same parameter estimates and associated standard errors (e.g. Table 2). This is impossible. Even if they repeated the experiment with the same experimental design, one would expect parameter estimates to vary within the range defined by their standard error. Using a different dataset for model fitting (with a different experiment design), should result in an even larger variation. Consequently, it is impossible the parameter estimates reported by the authors are the ones obtained when they follow the method for model fitting described in the M&M section.
## The article has little novelty
The main goal of the article is defined by the authors as “comparing omnibus modelling to conventional sequential first order / second order modelling” (25). However, this comparison is not novel. As stated by the authors in their response, they “have used omnibus modelling in predictive microbiology since 2013”. Moreover, this methodology (although with different names) has been used in plenty of scientific studies by groups from all over the world during the last 20 years (for instance, Fernández et al. (1999); den Besten et al. (2017); Garre et al. (2020); Cattani et al. (2016); Muramatsu et al. (2019); van Derlinden et al. (2008)), most of them comparing it to the two-step approach concluding that, in most cases, it is superior. Therefore, it is generally accepted by the scientific community that the one-step (or global, or omnibus) approach is superior, so the main goal of this research is to demonstrate something that is already known.
## The article is not properly contextualized
The authors claim in their rebuttal letter that “the term ‘omnibus’ is more specific and familiar to predictive microbiologists”. However, they also acknowledge that only Prof. Juneja and his collaborators (including the authors of this article) use that term. If it is so familiar, why is it only used by a single research group (and their collaborators)? One possible answer would be that fitting both the primary and secondary model in a single step is a methodology only applied by that group, something hinted by their claim that “USDA researchers and Gonzales-Barron and co-workers have been pioneers in omnibus modelling for predictive microbiology”. But this is untrue. This approach has been applied (at least) since the late 90s, although with other names (e.g. Fernández et al (1999) DOI: 10.1006/fmic.1999.0282). Nowadays, most research groups working in predictive microbiology are able to implement it, and there are software tools that use it (e.g. IPMP global fit, or bioinactivation). Consequently, the methodology is known well, but the “omnibus” term isn’t. The authors are free to use that term within the manuscript, but it should be properly described and it should be mentioned that other authors use the same methodology with a different name.
Besides that, it is not clear what modelling approaches are included within the “omnibus” term. The authors apply both “one-step fitting” and mixed-effects model. It is not clear whether “omnibus” includes both methods or just “one-step fitting”. This should be defined very clearly in the manuscript, especially considering that the main goal of the project is comparing this method against different ones.
Moreover, the authors do not justify the advantages of this approach with respect to two-step modelling in enough detail. They only state that (1) “there is no loss of information” and (2) “random effects can extract the variability in kinetic parameters that cannot be explained by the environmental conditions” (L79). The first statement is very open, as the term “information” can describe many different things. I believe they refer to the fact that most two-step models disregard the uncertainty/variability of the parameters of the primary model. But that is my interpretation of their statement, and an article should not leave so much to the reader’s interpretation. Regarding the second advantage, it is only true if the proper structure is defined for the mixed-effects model. A model based on “classical” regression (with only fixed effects) has a variance component to describe the residual variation. The advantage of mixed-effects models is that they can assign hypotheses to this variance, assigning it to different parts of the model. But this is only the case if the proper mixed-effects model is defined.
Apart from that, the authors totally disregard the disadvantages of the “omnibus” approach. For starters, its implementation and its analysis are much more complicated. The two-step approach can be applied using just Excel, whereas this approach requires statistical software. The parameters of a primary model can be “guessed” by just looking at the observed microbial count, but the parameters of the (Ratkowsky) secondary model cannot be guessed by simply looking at the graph. Also, the secondary model is “automatically” validated in the two-step approach, but that is not the case in the “omnibus”. For instance, although the authors claim in their response that “it is necessary to evaluate the ‘best’ primary and secondary models” and that this is not done “blindly”. However, unlike for the growth rate (e.g. Figure 2), it is not clear reading the manuscript how the secondary model for the lag phase has been defined or validated. So, based on the information provided in the manuscript, it seems this secondary model has been defined “blindly”.
## The experimental design does not allow a proper comparison between the methods (which is the main goal of the article)
It is hard to evaluate because the advantages of this approach are not properly presented in the manuscript. But, in my opinion, the experimental design does not allow a proper comparison between the “omnibus” and the “classical” method. According to the authors, one of the main advantages of this approach is a better description of variability as “random effects can extract the variability in kinetic parameters” (L79). This is further emphasized by making a couple of references to the variability in growth of L.mono at the end of the introduction (L85-L88). However, as acknowledged by the authors, their data does not allow to properly describe variability in the growth of this microorganism, as they only used 5 strains. If due to the narrow experimental design the data does not include information on this variability, how can it be used to prove that the “omnibus” approach can extract that variability better?
Apart from that, the authors are now fitting a model to data obtained at only two temperatures. How does that affect the comparison between the methods? Is the “omnibus” approach better for cases with few or many temperatures? This also raises questions regarding the validation of the secondary model. The authors claim that the secondary model is not applied “blindly” and is checked in the training set. How is this done if there are only two temperatures? Is the information in the validation set somehow used for model definition/training?
For these reasons, this work does not provide any additional information with respect to previous scientific studies comparing one-step and two-step modelling approaches.
Minor comments/responses to previous minor comments
---------------------------------------------------------------------------
- L30: “Indicated” instead of “indicatd”.
- L62-64: Kinetic parameters are usually estimated using laboratory media. Although there are some similarities in the methodology, this could be hardly considered a challenge test, as this methodology has a much more applied goal.
- L108: The information about what is a “biological replicate” should be included in the manuscript, as it affects the interpretation of the variance of the results.
- L118: Even if the effect on the model predictions is small, there will be an impact. If the authors are using Equation (4) as a simplification of Equation (3), they should use the same value for \alpha. Otherwise, it is not a simplification but a different model (even if the difference is small).
- L195: How do the authors conclude that a variance component “is not significant”?
- L211: Even if it is a “natural” covariance structure, it can introduce an overparameterization in the model that, due to parameter identifiability, can inflate the uncertainty of other parameters.
- L241: I agree that including them in the main body of the manuscript would be too much repetition. But, right now, there is no information to validate the primary models fitted for some conditions. This information should be included as supplementary material.
- L247: I have strong concerns about the estimates of Nmax. The authors claim the data has “enough curvature”, but Figure 3 shows the data has no or very little curvature.
- Table 1: Even if the software can calculate three significant digits, using 7 decimals for a growth rate in units of 1/h is information about microbial growth precise up to the order of milliseconds. Considering the available technologies, this is not unrealistic.
- Tables 2-6: The units of every model parameter should be included in the Table. The authors should not expect the reader to dive through the whole text to find out what are the units of, e.g., \beta or the variance components. This leads to confusion and misinterpretation.
Author Response
Major comments
----------------------
## It is impossible that a change in the training set does not affect the parameter estimates
In the original version of the manuscript, the authors used two replicates for model fitting and one for model validation. Based on my comments, they claim to have modified this method, using two temperatures for model fitting and one for model validation (L257-261). However, they report exactly the same parameter estimates and associated standard errors (e.g. Table 2). This is impossible. Even if they repeated the experiment with the same experimental design, one would expect parameter estimates to vary within the range defined by their standard error. Using a different dataset for model fitting (with a different experiment design), should result in an even larger variation. Consequently, it is impossible the parameter estimates reported by the authors are the ones obtained when they follow the method for model fitting described in the M&M section.
The data presented in tables 2 - 5 represent the fitting of the model to the complete data set. A separate validation exercise was carried out whereby the model was fitted to the data sets from two of the temperatures, 4.5 and 12 C and used to predict the growth at 7.8 oC for model validation (as described in lines 250 onwards). As the complete data set was used the values in the tables did not change from the original submission. An addition has been made to line 309 further clarifying that the complete data set was used.
## The article has little novelty
The main goal of the article is defined by the authors as “comparing omnibus modelling to conventional sequential first order / second order modelling” (25). However, this comparison is not novel. As stated by the authors in their response, they “have used omnibus modelling in predictive microbiology since 2013”. Moreover, this methodology (although with different names) has been used in plenty of scientific studies by groups from all over the world during the last 20 years (for instance, Fernández et al. (1999); den Besten et al. (2017); Garre et al. (2020); Cattani et al. (2016); Muramatsu et al. (2019); van Derlinden et al. (2008)), most of them comparing it to the two-step approach concluding that, in most cases, it is superior. Therefore, it is generally accepted by the scientific community that the one-step (or global, or omnibus) approach is superior, so the main goal of this research is to demonstrate something that is already known.
We addressed this point in our previous submission and are of the view that we have nothing further to add given that the reviewer is not making any suggestion for improvement.
## The article is not properly contextualized
The authors claim in their rebuttal letter that “the term ‘omnibus’ is more specific and familiar to predictive microbiologists”. However, they also acknowledge that only Prof. Juneja and his collaborators (including the authors of this article) use that term. If it is so familiar, why is it only used by a single research group (and their collaborators)? One possible answer would be that fitting both the primary and secondary model in a single step is a methodology only applied by that group, something hinted by their claim that “USDA researchers and Gonzales-Barron and co-workers have been pioneers in omnibus modelling for predictive microbiology”. But this is untrue. This approach has been applied (at least) since the late 90s, although with other names (e.g. Fernández et al (1999) DOI: 10.1006/fmic.1999.0282). Nowadays, most research groups working in predictive microbiology are able to implement it, and there are software tools that use it (e.g. IPMP global fit, or bioinactivation). Consequently, the methodology is known well, but the “omnibus” term isn’t. The authors are free to use that term within the manuscript, but it should be properly described and it should be mentioned that other authors use the same methodology with a different name.
In lines 78/79 in the introduction we have already made the point that this approach has several names. We have used omnibus to be consistent with previous work by this group and others. In addition we describe explicity in lines 155 onwards that omnibus modelling is is essentially a multilevel model, and was adjusted to each of the five L. monocytogenes strains datasets, as a non-linear mixed-effects regression.
Yes IPMP will allow a form of omnibus modelling but it does not allow the flexibility in the choice of fixed/random effects as undertaken and explained in this paper
Besides that, it is not clear what modelling approaches are included within the “omnibus” term. The authors apply both “one-step fitting” and mixed-effects model. It is not clear whether “omnibus” includes both methods or just “one-step fitting”. This should be defined very clearly in the manuscript, especially considering that the main goal of the project is comparing this method against different ones.
As above we we describe explicity in lines 155 onwards that omnibus modelling is is essentially a multilevel model, and was adjusted to each of the five L. monocytogenes strains datasets, as a non-linear mixed-effects regression. The appropriate equation are given from lines 155 - 166. in the section entitled omnibus modelling. A separate section entitled first order growth rate modelling is given from lines 116 onwards. We are of the view that the distinction between the two is already clearly laid out in the paper.
Moreover, the authors do not justify the advantages of this approach with respect to two-step modelling in enough detail. They only state that (1) “there is no loss of information” and (2) “random effects can extract the variability in kinetic parameters that cannot be explained by the environmental conditions” (L79). The first statement is very open, as the term “information” can describe many different things. I believe they refer to the fact that most two-step models disregard the uncertainty/variability of the parameters of the primary model. But that is my interpretation of their statement, and an article should not leave so much to the reader’s interpretation. Regarding the second advantage, it is only true if the proper structure is defined for the mixed-effects model. A model based on “classical” regression (with only fixed effects) has a variance component to describe the residual variation. The advantage of mixed-effects models is that they can assign hypotheses to this variance, assigning it to different parts of the model. But this is only the case if the proper mixed-effects model is defined.
We have amended the section starting line 81 to say that a two-step modelling approach undertaken as a mixed-effects regression is advantageous over the classical two-step modelling in the sense that (i) there is no loss of information about the uncertainty of the kinetic parameters of the primary model, and (ii) random effects can accommodate the variability in parameters that cannot be explained by the environmental conditions.
Apart from that, the authors totally disregard the disadvantages of the “omnibus” approach. For starters, its implementation and its analysis are much more complicated. The two-step approach can be applied using just Excel, whereas this approach requires statistical software. The parameters of a primary model can be “guessed” by just looking at the observed microbial count, but the parameters of the (Ratkowsky) secondary model cannot be guessed by simply looking at the graph. Also, the secondary model is “automatically” validated in the two-step approach, but that is not the case in the “omnibus”. For instance, although the authors claim in their response that “it is necessary to evaluate the ‘best’ primary and secondary models” and that this is not done “blindly”. However, unlike for the growth rate (e.g. Figure 2), it is not clear reading the manuscript how the secondary model for the lag phase has been defined or validated. So, based on the information provided in the manuscript, it seems this secondary model has been defined “blindly”.
We are unsure of the reviewers comments here. The reviewer has already stated above that "Nowadays, most research groups working in predictive microbiology are able to implement it, and there are software tools that use it (e.g. IPMP global fit, or bioinactivation). " but now the reviewer has concerns that the approach is complicated and can not be done in EXCEL. By clearly stating the mathematical approach in the paper, we are of the view that research groups working in predictive microbiology can use the approach presented in this paper.
In addition, the secondary model used for the lag phase is explicitly defined in equation 6.
## The experimental design does not allow a proper comparison between the methods (which is the main goal of the article)
It is hard to evaluate because the advantages of this approach are not properly presented in the manuscript. But, in my opinion, the experimental design does not allow a proper comparison between the “omnibus” and the “classical” method. According to the authors, one of the main advantages of this approach is a better description of variability as “random effects can extract the variability in kinetic parameters” (L79). This is further emphasized by making a couple of references to the variability in growth of L.mono at the end of the introduction (L85-L88). However, as acknowledged by the authors, their data does not allow to properly describe variability in the growth of this microorganism, as they only used 5 strains. If due to the narrow experimental design the data does not include information on this variability, how can it be used to prove that the “omnibus” approach can extract that variability better?
Five strains still have a variability which can be investigatesd using both methods. As stated in our first response, we never set out to conduct a major study on strain to strain variability. The focus of the paper was to demonstrate the method.
Apart from that, the authors are now fitting a model to data obtained at only two temperatures. How does that affect the comparison between the methods? Is the “omnibus” approach better for cases with few or many temperatures? This also raises questions regarding the validation of the secondary model. The authors claim that the secondary model is not applied “blindly” and is checked in the training set. How is this done if there are only two temperatures? Is the information in the validation set somehow used for model definition/training?
As stated above, the full model was developed using the full data set which had three temperatures. A separate validation exercise was carried out whereby the model was fitted to the data sets from two of the temperatures, 4.5 and 12 C and used to predict the growth at 7.8 oC for model validation (as described in lines 250 onwards). Interesting in the validation work, the model was able to predict pretty well the behaviour at 7.8 Deg C even though only two temperatures were used in the training set.
For these reasons, this work does not provide any additional information with respect to previous scientific studies comparing one-step and two-step modelling approaches.
For the reasons stated above and in the previous responses to reviewers, we would not accept this comment.
Minor comments/responses to previous minor comments
---------------------------------------------------------------------------
- L30: “Indicated” instead of “indicatd”.
Corrected
- L62-64: Kinetic parameters are usually estimated using laboratory media. Although there are some similarities in the methodology, this could be hardly considered a challenge test, as this methodology has a much more applied goal.
We are not sure of the reviewer's comment here, this was simply a statement in the introduction as background to why predictive models have been developed.
- L108: The information about what is a “biological replicate” should be included in the manuscript, as it affects the interpretation of the variance of the results.
As stated in the previous responses, For a biological replicate, the researcher went back to the original isolate stored at -20 Deg C and generated new starting inoculum. This has been now better explained in the text line 113.
- L118: Even if the effect on the model predictions is small, there will be an impact. If the authors are using Equation (4) as a simplification of Equation (3), they should use the same value for \alpha. Otherwise, it is not a simplification but a different model (even if the difference is small).
We are simply using the appropriate values of alpha reported previously by Huang [10] for the reduced model compared to the full model as described by Huang [10]
- L195: How do the authors conclude that a variance component “is not significant”?
It was the standard deviation of the random effects that was shown to be non-significant consistently (line 200)
- L211: Even if it is a “natural” covariance structure, it can introduce an overparameterization in the model that, due to parameter identifiability, can inflate the uncertainty of other parameters.
We do not use the word "natural" covariance structure anywhere in the manuscript. If there is potential for covariance, it should be allowed for in the model as we do through introducing covaraance matrices.
- L241: I agree that including them in the main body of the manuscript would be too much repetition. But, right now, there is no information to validate the primary models fitted for some conditions. This information should be included as supplementary material.
There is a data availability statement already included which states that the data presented in this study are available on request from the corresponding author.
- L247: I have strong concerns about the estimates of Nmax. The authors claim the data has “enough curvature”, but Figure 3 shows the data has no or very little curvature.
This is one advantage of the omnibus approach as it is estimating the complete shape of the curve based on the other two temperatures where stationary phase was achieved.
- Table 1: Even if the software can calculate three significant digits, using 7 decimals for a growth rate in units of 1/h is information about microbial growth precise up to the order of milliseconds. Considering the available technologies, this is not unrealistic.
We have amended table 1 to two significant digits
- Tables 2-6: The units of every model parameter should be included in the Table. The authors should not expect the reader to dive through the whole text to find out what are the units of, e.g., \beta or the variance components. This leads to confusion and misinterpretation.
The units have been added to the tables
Reviewer 4 Report
The concerns of the present reviewer regarding the scientific soundness of the present manuscript remain. The comments raised in the first review round were not satisfactorily addressed by the authors.
Author Response
The concerns of the present reviewer regarding the scientific soundness of the present manuscript remain. The comments raised in the first review round were not satisfactorily addressed by the authors.
We are disappointed that the reviewer continues to have reservations relating to this paper. We are of the view that we gave a comprehensive response in the first round to the reviewers suggestions. Unfortunately as the reviewer does not give any further suggestions for change, we can not respond further to the comment above beyond our original response. We note however that two reviewers have responded positively to the paper and recommend publication.